# LAPRAS: Learning-Augmented PRivate Answering for linear query Streams

**Pranay Mundra** [1]  **Adam Sealfon** [2]  **Ziteng Sun** [2]  **Quanquan C. Liu** [1]

## Abstract

Modern database workloads are highly predictable: query streams are dominated by recurring jobs and templates, even when their arrival order is not known in advance. This motivates a learning-augmented view of online differentially private (DP) analytics: can algorithms utilize predictions about *which* queries will occur to improve utility under a single global privacy budget, while remaining robust when predictions are wrong? We study online DP query answering, where a curator must answer a stream $Q$ of $S$ linear queries arriving in uniformly random order under privacy budget $(\varepsilon, \delta)$. We present *LAPRAS*, which assumes access to an oracle that outputs a prediction set of queries likely to appear in the stream and uses it to guide privacy spending. LAPRAS answers predicted queries using the offline-optimal Matrix Mechanism and answers the remaining queries online from a residual budget. To pace spending across an unknown number of unpredicted queries, we introduce *Smooth Allocation*, which forms an unbiased stopping-time estimate $\widehat{B}$ from the first $T = \Theta(\log^2 S)$ unpredicted queries and continuously recalibrates per-query expenditure. Empirically, over two real datasets, we validate the intended consistency–robustness trade-off: LAPRAS achieves near-offline utility under high overlap and degrades gracefully to baseline-level performance when overlap is low.

## 1. Introduction

In the era of data-driven decision-making, organizations collect vast volumes of sensitive information, ranging from financial transaction logs to medical health records. The utility of this data lies in the ability to run aggregate analytics–counting queries, histograms, and linear summations–that reveal population level trends without compromising individual privacy. **Differential Privacy (DP)** (Dwork & Roth, 2014) has emerged as the rigorous standard for such systems, quantifying privacy loss and ensuring that the output of a computation remains virtually indistinguishable whether any single individual's record is included or excluded.

However, deploying differential privacy (DP) in real-world database systems exposes a fundamental mismatch between *online query answering* and the *offline* assumptions underlying optimal DP mechanisms. In the offline model, the curator is given a fixed workload W in advance, enabling global optimization of noise: mechanisms such as the Matrix Mechanism (Li et al., 2010; McKenna et al., 2020) exploit correlations in W to design a strategy that minimizes total error. Modern systems that support real-time dashboards, monitoring, and interactive exploration instead face a stream of queries $q_1, \ldots, q_S$ and must return each private answer immediately, without knowledge of future queries. This uncertainty provably separates the online and offline regimes: there exist workloads for which any online mechanism incurs exponentially larger error than an offline mechanism that sees the full workload (Bun et al., 2019; Hardt & Rothblum, 2010). Intuitively, without knowing whether future queries will be correlated, an online mechanism must budget conservatively, often adding noise that render the data useless.

Real-world database workloads are highly predictable. Large-scale studies show that production query streams are dominated by recurring jobs and templates: over $60\%$ of SCOPE jobs recur on fixed schedules (Jindal et al., 2018; Wu et al., 2024), and a small set of templates can account for over $90\%$ of resource consumption in SQL Server and Azure SQL telemetry (Zhang et al., 2018). This structure is already leveraged by self-driving components such as SageDB (Kraska et al., 2019) and Sibyl (Huang et al., 2024), which learn workload regularities to anticipate query characteristics and optimize execution. Motivated by this evidence, we assume access to predictions about *which* queries are likely to appear, while making only a weak assumption about *when* they appear by modeling the stream as a uniformly random order of the realized workload. This is a natural abstraction for systems in which multiple recurring

[1]Yale University, New Haven, CT, USA [2]Google Research, New York City, NY, USA. Correspondence to: Pranay Mundra <pranay.mundra@yale.edu>, Adam Sealfon <adamsealfon@google.com>, Ziteng Sun <zitengsun@google.com>, Quanquan C. Liu <quanquan.liu@yale.edu>.

*Proceedings of the $43^{rd}$ International Conference on Machine Learning*, Seoul, South Korea. PMLR 306, 2026. Copyright 2026 by the author(s).

jobs, user interactions, and scheduled pipelines interleave, so the workload structure is stable but the precise arrival order is not reliably known in advance. This setting naturally aligns with **learning-augmented algorithms** (Khodak et al., 2023): predictions can be exploited to achieve near-optimal utility when accurate (consistency), while the mechanism must degrade gracefully to standard worst-case guarantees when they are not (robustness).

We propose **LAPRAS**, a framework that uses workload predictions obtained, for example, from learned models over prior query logs to improve online differentially private query answering. Given a predictor that identifies a subset P of queries likely to appear, we answer P using an offline-optimal batch mechanism (e.g., the Matrix Mechanism (Li et al., 2010)), exploiting correlations to precompute low-noise releases that can be served at zero additional privacy cost when they arrive. The remaining queries must be answered online from a residual budget, creating a budget-pacing problem: how do we allocate the remaining budget across an unknown number of unpredicted queries? LAPRAS addresses this with *Smooth Allocation*: using the random-order assumption, we form an unbiased stopping-time estimator $\hat{B}$ from the arrival positions of the first $T = \Theta(\log^2 S)$ unpredicted queries, and allocate per-query privacy spend proportional to $\varepsilon_{\mathrm{rem}}/\hat{B}$, updating as $\hat{B}$ stabilizes over the stream. Our contributions are:

- **LAPRAS:** We formalize a learning-augmented DP mechanism that exploits the power of predictions, with batch processing for predicted queries and adaptive online processing for unpredicted ones. This design enables the system to smoothly interpolate between the utility of the best offline algorithms (in high-overlap/accurate prediction regimes) and robust online baselines (in low-overlap regimes).

- **Smooth Allocation & Unbiased Estimation:** We derive and analyze the Smooth Allocation strategy. We provide the first proof that a stopping-time estimator $\hat{B}$ based on bad query arrival positions is unbiased and concentrates sufficiently fast to drive privacy budget allocation without violating composition guarantees.

- **Theoretical Utility Guarantee:** For a stream $\mathcal{S}$ with $B$ unpredicted queries, we prove that the total expected squared error of LAPRAS satisfies $\sum_{q \in \mathcal{S}} \mathbb{E}[U_{\mathrm{LAPRAS}}(q)^2] = O\left(\frac{B^2 \ln(1/\delta)}{\varepsilon^2}\right) + O\left(\sum_{q \in \mathcal{S}} \mathbb{E}[U_{\mathrm{MM}}(q)^2]\right)$, and moreover $\sum_{q \in \mathcal{S}} \mathbb{E}[U_{\mathrm{LAPRAS}}(q)^2] \leq c \cdot \sum_{q \in \mathcal{S}} \mathbb{E}[U_{Online}(q)^2]$ for a fixed constant $c \geq 1$.

- **Empirical Validation:** We evaluate on two datasets against OfflineMM and the online Independent Noise baseline. LAPRAS exhibits the intended consistency–robustness trade-off: at high overlap ($\rho \approx 1$) it reduces median MAE by over an order of magnitude (Adult:

193.4 → 14.3; Gowalla: 181.2 → 17.1, $\varepsilon{=}1.0$), while at low overlap ($\rho \approx 0$) it remains comparable to Independent Noise (Adult: 201.8 vs. 186.5; Gowalla: 213.9 vs. 204.1).

## 2. Related Works

The hardness of answering adaptive, dynamic query streams under differential privacy is well established (Bun et al., 2019; Hardt & Rothblum, 2010). Classical mechanisms such as Private Multiplicative Weights (PMW) (Hardt & Rothblum, 2010) maintain a synthetic database to answer queries with provable error guarantees, but their update steps scale poorly with the domain size, limiting practicality in high-dimensional settings. LAPRAS targets a different point in the design space: it retains a simple computational profile (matrix operations and per-query noise) while addressing the core online bottleneck: *how to spend a fixed global budget when the number of costly queries is unknown*. Privacy odometers and filters (Whitehouse et al., 2023; Rogers et al., 2016) provide accounting primitives for adaptive composition, but they are descriptive rather than prescriptive: they track privacy loss, whereas LAPRAS's stopping-time estimator and Smooth Allocation provide an explicit *spending policy* that allocates the remaining budget over the residual stream to improve utility.

A complementary line of work reduces online privacy cost by reusing previously released noisy answers. CacheDP (Mazmudar et al., 2022) is representative: it maintains a DP cache and answers new queries via post-processing when they can be expressed using cached information, thereby reducing the need for fresh noise. This approach is fundamentally *reactive* and depends on historical redundancy (with an inherent cold-start cost), whereas LAPRAS is *proactive*: it uses predictions to precompute a workload-aware representation before the stream arrives and then paces spending only on the unpredicted remainder. For this reason, a direct head-to-head experimental comparison is not especially informative: CacheDP is designed to minimize *additional privacy cost* subject to a target accuracy requirement using a cache-dependent state, while our setting fixes a global budget and studies the resulting error as a function of prediction accuracy. In practice, these approaches are best viewed as orthogonal: CacheDP exploits repetition in past queries, while LAPRAS exploits predictability of future workload structure.

## 3. Privacy Tools

Our work is based on the framework of Differential Privacy (DP) (Dwork & Roth, 2014), which provides strong, worst-case guarantees against privacy leakage.

**Definition 3.1** (($\varepsilon, \delta$)-Differential Privacy (Dwork &

Roth, 2014)). *A randomized algorithm $\mathcal{A}$ satisfies $(\varepsilon, \delta)$-**Differential Privacy** if for any two adjacent databases $D_1$ and $D_2$ that differ by at most one record, and for any set of possible outputs $O \subseteq Range(\mathcal{A})$, the following inequality holds:*

$$P[\mathcal{A}(D_1) \in O] \le e^{\varepsilon} P[\mathcal{A}(D_2) \in O] + \delta$$

Here, $\varepsilon$ (the privacy budget) bounds the multiplicative divergence between the output distributions, controlling the worst-case information leakage. The parameter $\delta$ (the failure probability) represents the probability that the multiplicative bound fails to hold. Typically, $\delta$ is chosen to be negligible (e.g., $\delta < 1/|D|$). If an algorithm satisfies the definition above for $\delta = 0$, then it is $\varepsilon$-differentially private.

**Definition 3.2** (Global Sensitivity (Dwork & Roth, 2014)). *For a function $f : \mathcal{D} \to \mathbb{R}^k$, its $L_p$ global sensitivity, denoted $\Delta_p f$, is the maximum possible change in the output of $f$ over all pairs of adjacent databases:*

$$\Delta_p f = \max_{D_1, D_2} ||f(D_1) - f(D_2)||_p$$

In this work, we focus on linear counting queries where adding or removing a single individual's data changes the count by at most 1. Therefore, the $L_1$ and $L_2$ global sensitivities are both 1.

**Theorem 3.3** (The Gaussian Mechanism (Dwork & Roth, 2014)). *Let $f : \mathcal{D} \to \mathbb{R}^k$ be a function with $L_2$ global sensitivity $\Delta_2 f$. The algorithm $\mathcal{A}(D) = f(D) + N(0, \sigma^2 I_k)$, which adds noise from a Gaussian distribution with variance $\sigma^2$ to each component of the output, is $(\varepsilon, \delta)$-differentially private for $\varepsilon \in (0, 1)$ if the standard deviation $\sigma$ is chosen such that:*

$$\sigma = \frac{\Delta_2 f \sqrt{2 \ln(1.25/\delta)}}{\varepsilon}$$

**Theorem 3.4** (The Analytic Gaussian Mechanism (Balle & Wang, 2018)). *The classical bound above is loose, particularly in the high-privacy regime (small $\varepsilon$). (Balle & Wang, 2018) derived the necessary and sufficient conditions for Gaussian noise to satisfy DP using the exact cumulative distribution function (CDF) of the normal distribution, $\Phi$. A Gaussian mechanism with noise scale $\sigma$ is $(\varepsilon, \delta)$-DP if and only if:*

$$\Phi\left(\frac{\Delta_2}{2\sigma} - \frac{\varepsilon\sigma}{\Delta_2}\right) - e^{\varepsilon}\Phi\left(-\frac{\Delta_2}{2\sigma} - \frac{\varepsilon\sigma}{\Delta_2}\right) \le \delta$$

LAPRAS employs this Analytic Gaussian Mechanism (AGM). By numerically inverting this inequality, we can find the smallest possible $\sigma$ for a given $(\varepsilon, \delta)$, thereby maximizing utility (minimizing error) for a fixed privacy budget.

**Theorem 3.5** (Matrix Mechanism (Li et al., 2010)). *Given an $m \times n$ workload matrix $W$, a $p \times n$ strategy matrix*

$\mathbf{A}$ *that supports $W$ and a differentially private algorithm $\mathcal{K}(\mathbf{A}, x)$ that answers $\mathbf{A}$ with a given database $x$. The matrix mechanism $\mathcal{M}_{\mathcal{K}, \mathbf{A}}$ outputs the following vector:*

$$\mathcal{M}_{\mathcal{K}, \mathbf{A}} = W \mathbf{A}^{+} \mathcal{K}(\mathbf{A}, x)$$

The mechanism selects the optimal strategy matrix $\mathbf{A}$ by solving a convex optimization problem, typically formulated as a Semidefinite Program (SDP), which minimizes the variance of the reconstructed answers subject to a constraint on the sensitivity of $\mathbf{A}$, and derives estimates for the original workload using the Moore-Penrose pseudoinverse $\mathbf{A}^{+}$.

**Theorem 3.6** (Basic Composition (Dwork & Roth, 2014)). *Let $\mathcal{M}_1, \mathcal{M}_2, \ldots, \mathcal{M}_k : \mathcal{X}^n \to \mathcal{Y}$ be randomized algorithms. Suppose $\mathcal{M}_j$ is $(\varepsilon_j, \delta_j)$-DP for each $j \in [k]$. Define $\mathcal{M} : \mathcal{X}^n \to \mathcal{Y}$ by $\mathcal{M}(x) = (\mathcal{M}_1(x), \mathcal{M}_2(x), \ldots, \mathcal{M}_k(x))$, where each algorithm is run independently. Then $\mathcal{M}$ is $(\varepsilon, \delta)$-DP where $\varepsilon = \sum_{j=1}^{k} \varepsilon_j$, and $\delta = \sum_{j=1}^{k} \delta_j$.*

**Theorem 3.7** (Post-Processing Immunity (Dwork & Roth, 2014)). *Let $\mathcal{A}$ be an $(\varepsilon, \delta)$-differentially private algorithm. Let $g$ be an arbitrary randomized or deterministic function that takes the output of $\mathcal{A}$ as input. The algorithm $g(\mathcal{A}(D))$ is also $(\varepsilon, \delta)$-differentially private.*

**Definition 3.8.** *[Private Online Query Aswering] Let $x \in \mathbb{R}^n$ be a private data vector (e.g., derived from a database of size $N$), and let $Q = (q_1, \ldots, q_S)$ be a stream of $S$ linear counting queries arriving in a uniformly random order. An untrusted oracle, based on prior logs or learned workload structure, outputs a prediction set $P$ that may overlap with the stream. The goal is to design an online mechanism that, upon receiving each $q_t$, outputs an answer $a_t$ immediately and uses $P$ to adapt its per-query privacy spending $(\varepsilon_t, \delta_t)$ so that the overall interaction is $(\varepsilon, \delta)$-differentially private, while minimizing the resulting error of the released answers (e.g., MAE) relative to prediction-oblivious online baselines.*

## 4. LAPRAS

We present **LAPRAS**, a learning-augmented framework that tackles the problem of private online query answering (Definition 3.8) by using the oracle's prediction set $P$ to guide privacy spending. Given $P$, LAPRAS classifies each arriving query $q_t$ as *good* if $q_t \in P$ and *bad* otherwise; we write $G := |\{t : q_t \in P\}|$ and $B := S - G$ for the number of good and bad queries, respectively. At a high level, LAPRAS handles the good portion with a workload-aware release that exploits correlations among predicted queries (Matrix Mechanism (Li et al., 2010)), producing low-noise information that can be served when good queries occur with no additional privacy cost at query time. Bad queries, in contrast, must be answered online by adding independent noise, drawing from a residual budget.

**Algorithm 1** LAPRAS

1: **Input:** Query set $Q$ of size $S$, global privacy budget $\varepsilon, \delta$, budget strategy *strat*, *isSmooth*, $x$ database
2: $T \leftarrow \lceil \log^2 S \rceil$, $\delta_i \leftarrow \frac{\delta}{S+1}$
3: **if** *isSmooth* **then**
4: $\quad \varepsilon_{\text{MM}}, \varepsilon_{\text{bad}}, \varepsilon_{\text{reserve}} \leftarrow$ splitBudget $(\varepsilon, strat)$
5: **else**
6: $\quad \varepsilon_{\text{MM}}, \varepsilon_{\text{badInit}}, \varepsilon_{\text{remBad}}, \varepsilon_{\text{reserve}} \leftarrow$ splitBudget $(\varepsilon, strat)$
7: **end if**
8: Define minimal reserve threshold $\varepsilon_{\min} > 0$
9: $P \leftarrow$ Oracle$(Q)$    *// Predicted good queries*
10: $A \leftarrow$ MatrixMechanism$(P, x, \varepsilon_{\text{MM}}, \delta_i)$
11: $b \leftarrow 0$,   $n \leftarrow 0$,   $B_{\text{est}} \leftarrow 0$,   $\hat{B} \leftarrow 0$,   remBad $\leftarrow 0$
12: **for** each query $q \in Q$ **do**
13: $\quad n \leftarrow n + 1$    *// Total queries seen*
14: $\quad$ **if** $q \in P$ **then**
15: $\qquad ans[n] \leftarrow A(q)$
16: $\quad$ **else**
17: $\qquad b \leftarrow b + 1$
18: $\qquad$ **if** $b \leq T$ **then**
19: $\qquad\quad$ **if** $b = 1$ **then**
20: $\qquad\qquad \hat{B} \leftarrow S - n + 1$
21: $\qquad\quad$ **else**
22: $\qquad\qquad \hat{B} \leftarrow S \cdot \frac{(b-1)}{(n-1)}$
23: $\qquad\quad$ **end if**
24: $\qquad\quad$ **if** *isSmooth* **then**
25: $\qquad\qquad \varepsilon_b \leftarrow \frac{\varepsilon_{\text{bad}}}{\max(1, \hat{B} - b) + 1}$
26: $\qquad\qquad ans[n] \leftarrow$ AGM$\big(q, x, \varepsilon_b, \delta_i\big)$
27: $\qquad\qquad \varepsilon_{\text{bad}} \leftarrow \varepsilon_{\text{bad}} - \varepsilon_b$
28: $\qquad\quad$ **else**
29: $\qquad\qquad$ Reallocate$(\varepsilon_{\text{reserve}}, \varepsilon_{\text{initBad}})$
30: $\qquad\qquad ans[n] \leftarrow$ AGM$\big(q, x, \frac{\varepsilon_{\text{initBad}}}{T}, \delta_i\big)$
31: $\qquad\qquad \varepsilon_{\text{badInit}} \leftarrow \varepsilon_{\text{badInit}} - \big(\frac{\varepsilon_{\text{badInit}}}{T}\big)$
32: $\qquad\quad$ **end if**
33: $\qquad\quad$ **if** $b = T$ **then**
34: $\qquad\qquad B_{\text{est}} \leftarrow S \cdot \frac{(T-1)}{(n-1)}$    *// Estimate total bad queries*
35: $\qquad\qquad \hat{B} \leftarrow B_{\text{est}}$
36: $\qquad\qquad$ remBad $\leftarrow \max(B_{\text{est}} - T, 1)$
37: $\qquad\qquad$ **if** not *isSmooth* **then**
38: $\qquad\qquad\quad \varepsilon_{\text{remBad}} \mathrel{+}= \varepsilon_{\text{badInit}}$    *// Add any left over budget*
39: $\qquad\qquad$ **end if**
40: $\qquad\quad$ **end if**
41: $\qquad$ **else**
42: $\qquad\quad$ **if** $b \leq B_{\text{est}}$ **then**
43: $\qquad\qquad$ **if** *isSmooth* **then**
44: $\qquad\qquad\quad \varepsilon_b \leftarrow \frac{\varepsilon_{\text{bad}}}{\max(1, \hat{B} - b) + 1}$
45: $\qquad\qquad\quad ans[n] \leftarrow$ AGM$\big(q, x, \varepsilon_b, \delta_i\big)$
46: $\qquad\qquad\quad \varepsilon_{\text{bad}} \leftarrow \varepsilon_{\text{bad}} - \varepsilon_b$
47: $\qquad\qquad$ **else**
48: $\qquad\qquad\quad ans[n] \leftarrow$ AGM$\big(q, x, \frac{\varepsilon_{\text{remBad}}}{\text{remBad}}, \delta_i\big)$
49: $\qquad\qquad\quad \varepsilon_{\text{remBad}} \leftarrow \varepsilon_{\text{remBad}} - \big(\frac{\varepsilon_{\text{remBad}}}{\text{remBad}}\big)$
50: $\qquad\qquad$ **end if**
51: $\qquad\quad$ **else**
52: $\qquad\qquad$ **if** $\varepsilon_{\text{reserve}} < \varepsilon_{\min}$ **then**
53: $\qquad\qquad\quad$ **Stop answering further bad queries.**
54: $\qquad\qquad\quad$ **break**
55: $\qquad\qquad$ **else**
56: $\qquad\qquad\quad ans[n] \leftarrow$ AGM$\big(q, x, \frac{\varepsilon_{\text{reserve}}}{2}, \delta_i\big)$
57: $\qquad\qquad\quad \varepsilon_{\text{reserve}} \leftarrow \frac{\varepsilon_{\text{reserve}}}{2}$
58: $\qquad\qquad$ **end if**
59: $\qquad\quad$ **end if**
60: $\qquad$ **end if**
61: $\quad$ **Output** $ans$
62: $\quad$ **end if**
63: **end for**

The main challenge is **budget pacing**: the mechanism must allocate a finite residual budget across an *unknown* number of bad queries $B$. LAPRAS addresses this using the random-order assumption. Let $T = \lceil \log^2 S \rceil$. When the $T$-th bad query appears at position $n$ in the stream, we form the stopping-time estimator $B_{\text{est}} = S \cdot \frac{T-1}{n-1}$, which is unbiased for $B$. This estimate drives our allocation strategies, allowing LAPRAS to calibrate per-query noise to the realized number of bad queries rather than budgeting pessimistically against $S$. We give the full pseudocode in Algorithm 1.

### 4.1. Privacy Budget Allocation

We partition the global budget $\varepsilon$ into four components: $\varepsilon_{\text{MM}}$ for the matrix mechanism on the predicted set; $\varepsilon_{\text{badInit}}$ for the warm-up phase of bad queries; $\varepsilon_{\text{remBad}}$ for the remaining bad-query stream; and $\varepsilon_{\text{reserve}}$ as a safety buffer. We also define a threshold $\varepsilon_{\min} > 0$; if $\varepsilon_{\text{reserve}} < \varepsilon_{\min}$, the mechanism halts to avoid privacy violations. We evaluate four allocation strategies (Table 1) to study trade-offs between predicted-query accuracy, pacing on bad queries, and overall robustness, clarifying how the budget split shapes the privacy–utility trade-off. We set $\delta_i = \delta/(S+1)$.

*Table 1.* Budget split strategies used in our experiments.

| Name | $\varepsilon_{\text{MM}}$ | $\varepsilon_{\text{badInit}}$ | $\varepsilon_{\text{bad}}$ | $\varepsilon_{\text{reserve}}$ |
|---|---|---|---|---|
| equal | $\varepsilon/4$ | $\varepsilon/4$ | $\varepsilon/4$ | $\varepsilon/4$ |
| matrix-heavy | $\varepsilon/2$ | $\varepsilon/6$ | $\varepsilon/6$ | $\varepsilon/6$ |
| query-heavy | $\varepsilon/6$ | $\varepsilon/3$ | $\varepsilon/3$ | $\varepsilon/6$ |
| reserve-heavy | $\varepsilon/6$ | $\varepsilon/6$ | $\varepsilon/6$ | $\varepsilon/2$ |

#### 4.1.1. STATIC ALLOCATION

We apply the Matrix Mechanism (Li et al., 2010) to the oracle's predicted set P using budget $\varepsilon_{\text{MM}}$ and store these pre-computed answers. For the first $T = \lceil \log^2 S \rceil$ bad queries ($b \leq T$), we answer with independent noise using per-query budget $\varepsilon_{\text{badInit}}/T$, expending $\varepsilon_{\text{badInit}}$ in total. At $b = T$, we compute an unbiased streaming estimate $B_{\text{est}}$ of the total number of bad queries; thereafter, each bad query with $b \leq B_{\text{est}}$ is answered using $\varepsilon_{\text{bad}}/(B_{\text{est}} - T)$, expending $\varepsilon_{\text{bad}}$ overall. If $b > B_{\text{est}}$, we draw from a reserve budget $\varepsilon_{\text{reserve}}$ that is halved after each use and stop answering once it falls below $\varepsilon_{\min}$. While robust, Static Allocation is rigid; it commits to a single density estimate early in the stream, leading to suboptimal utility if the initial distribution of bad queries is an anomaly (e.g., a dense cluster), or if we have $B < T$ bad queries, we never have an estimate and thus waste a significant portion of the budget.

#### 4.1.2. SMOOTH ALLOCATION

To mitigate the rigidity of the static approach, we introduce Smooth Allocation, a control-theoretic method that continuously recalibrates the privacy spend. Rather than dividing the budget, this strategy combines it into a single active

pool, $\varepsilon_{\text{pool}} = \varepsilon_{\text{badInit}} + \varepsilon_{\text{remBad}}$. The core idea is to treat the estimation of $B$ as a dynamic process that improves as more of the stream is observed. Let $n_b$ denote the position in the stream where the $b$-th bad query arrives. For every bad query $b \geq 2$, we compute an instantaneous estimate $\widehat{B}(b) = S \cdot \frac{b-1}{n_b-1}$. When answering the $b$-th bad query, the mechanism utilizes the estimate, $\widehat{B}(b)$, to project the number of remaining bad queries: $\widehat{B}_{\text{rem},b} = \max(1, \widehat{B}(b) - b)$. The privacy budget $\varepsilon_b$ for the current query is then derived by equidistributing the remaining pool $\varepsilon_{\text{rem},b-1}$ over this projection:

$$\varepsilon_b = \frac{\varepsilon_{\text{rem},b-1}}{\widehat{B}_{\text{rem},b} + 1}, \quad \text{where} \quad \varepsilon_{\text{rem},b} = \varepsilon_{\text{rem},b-1} - \varepsilon_b.$$

The $+1$ term regularizes early updates to avoid overspending. If bad queries are sparse, $\varepsilon_b$ increases; if dense, $\varepsilon_b$ decreases. After the $T$-th bad query we fix an unbiased estimate and use it for pacing, while $\varepsilon_{\text{reserve}}$ preserves worst-case robustness.

## 4.2. Theoretical Analysis

We defer some proofs to the Appendix A due to space constraints.

**Definition 4.1** (Negative Hypergeometric Distribution)**.** *Let $N$ be the size of a finite population containing exactly $K$ successes and the $N - K$ failures. Suppose we draw items without replacement, until $r$ failures are encountered. Define the random variable*

$$Y = \text{the number of successes until we see } r \text{ failures}$$

*Then $Y$ is said to have a negative hypergeometric distribution with parameters $(N, K, r)$, denoted by*

$$Y \sim \text{NHG}\,(N, K, r)$$

**Theorem 4.2.** *Given a stream of $S$ queries, in random order, out of which exactly $B \geq T$ are bad, and $G = S - B$ are good. Fix $T = \lceil \log^2(S) \rceil$, and let $L$ be the number of queries until we see $T$ bad queries. Then, we get an unbiased estimator for the number of bad queries:*

$$\hat{B} = \frac{S\,(T-1)}{L-1},$$

*where*

$$\mathbf{E}\big[\hat{B}\big] = B$$

*Proof.* Let $Y$ be a random variable that denotes the number of good queries in the first $L$ queries, i.e., the number of good queries before we see $T$ bad queries. Then since it's a random order stream, we have that:

$$Y \sim \text{NHG}\big(S, G, T\big),$$

We have that:

$$L = Y + T$$

Now, we want to show that $\mathbb{E}\big[\hat{B}\big] = B$. Let

$$D = \frac{T-1}{L-1} = \frac{T-1}{Y+T-1}$$

Now, using Lemma A.1 (for $k = 1$), we have $\mathbb{E}\big[D\big] = \frac{B}{S}$: Using this, we have the following:

$$\mathbb{E}\big[\hat{B}\big] = S \times \mathbb{E}\big[\frac{T-1}{L-1}\big] = S \times \frac{B}{S} = B$$

$\square$

**Theorem 4.3.** *For a fixed integer $T \geq 3$, the variance of the estimator $\hat{B}$ is bounded by:*

$$\text{Var}[\hat{B}] < \frac{S(T-1)}{(S-1)(T-2)}B(B-1) - B^2$$

*where $B$ is the number of bad queries and $S$ is the total number of queries.*

**Corollary 4.4.** *With $T = \lceil \log^2(S) \rceil$, the variance of the estimator $\hat{B}$ has the following asymptotic upper bound:*

$$\text{Var}[\hat{B}] = O\left(\frac{B^2}{\log^2(S)}\right)$$

**Lemma 4.5.** *Let $\varepsilon_{\text{pool}}$ be the initial budget allocated to the Smooth Allocation strategy and $B$ be the total count of bad queries. The total budget expended satisfies:*

$$\sum_{i=2}^{B} \varepsilon_i < \varepsilon_{\text{pool}} \tag{1}$$

**Theorem 4.6** (Privacy Guarantee)**.** *The described algorithm is $(\varepsilon, \delta)$-differentially private.*

*Proof.* We apply post-processing and basic composition for $(\varepsilon, \delta)$-DP.

**Budget split.** The algorithm partitions the total privacy budget as $\varepsilon = \varepsilon_{\text{MM}} + \varepsilon_{\text{pool}} + \varepsilon_{\text{reserve}}$.

**Per-release $\delta$ accounting.** Let $K$ upper bound the number of randomized database accesses. LAPRAS answers a stream of $S$ queries with at most one randomized release per query, and performs one additional offline release for the predicted set; thus $K = S + 1$. We set $\delta_i := \frac{\delta}{K} = \frac{\delta}{S+1}$. Every database access in the algorithm is implemented as an $(\varepsilon_j, \delta_i)$-DP mechanism (with the appropriate $\varepsilon_j$ drawn from the corresponding budget component).

**Precomputation phase.** The algorithm runs the Matrix Mechanism on the predicted workload $P$ with parameters $(\varepsilon_{\text{MM}}, \delta_i)$, incurring privacy loss $(\varepsilon_{\text{MM}}, \delta_i)$.

**Unpredicted-query phase.** Each unpredicted query answer is produced by adding independent noise calibrated to $(\varepsilon_i, \delta_i)$. For Static Allocation, the total $\varepsilon$ spent across these answers is exactly $\varepsilon_{\text{badInit}} + \varepsilon_{\text{bad}} = \varepsilon_{\text{pool}}$. For Smooth Allocation, by budget soundness, $\sum_i \varepsilon_i \leq \varepsilon_{\text{pool}}$. If overflow occurs, additional answers are produced using per-query parameters $(\varepsilon_i, \delta_i)$ drawn from the reserve. The geometric decay rule ensures $\sum_i \varepsilon_i \leq \varepsilon_{\text{reserve}}$. In either case, the total $\varepsilon$ spent in this phase is at most $\varepsilon_{\text{pool}}$, and the number of releases is at most $S$, so the composed privacy loss is at most $(\varepsilon_{\text{pool}} + \varepsilon_{\text{reserve}}, S\delta_i)$.

**Post-processing.** All other computations (membership checks in $P$, the estimator $\widehat{B}$, and budget updates) depend only on public information and previously released DP outputs. By post-processing, they incur no additional privacy loss.

**Composition.** By basic composition over the precomputation release and at most $S$ online releases, the overall privacy loss is

$$(\varepsilon_{\text{MM}} + \varepsilon_{\text{pool}} + \varepsilon_{\text{reserve}}, \ \delta_0 + S\delta_0) = (\varepsilon, \ (S+1)\delta_0)$$
$$= (\varepsilon, \delta)$$

Therefore, the algorithm satisfies $(\varepsilon, \delta)$-differential privacy. $\square$

Next we give the theoretical utility guarantee of our algorithm in terms of the number of bad queries $B$ in the input stream and the utility of the offline Matrix Mechanism algorithm given all of the queries in the stream and the online naive independent noise algorithm. For any query $q$, let $U_{\text{MM}}(q) = |\hat{a} - a|$ be a random variable indicating absolute error of the Matrix Mechanism on $q$, where $a$ is the true answer and $\hat{a}$ is the released (noisy) answer. Define $U_O(q)$ and $U_{\text{LAPRAS}}(q)$ analogously as the errors of the online baseline and LAPRAS, respectively.

**Theorem 4.7** (Utility Guarantee). *Given a total of $M$ queries and $B$ bad queries in the stream $\mathcal{S}$ and where $P \subseteq \mathcal{S}$, for any query $q \in \mathcal{S}$, it holds that*

$$\sum_{q \in \mathcal{S}} \mathbb{E}[U_{LAPRAS}(q)^2] \leq \frac{cB^3 \ln(1/\delta)}{\varepsilon^2} + c \sum_{q \in \mathcal{S}} \mathbb{E}[U_{MM}(q)^2]$$
$$\leq d \cdot \sum_{q \in \mathcal{S}} \mathbb{E}[U_O(q)^2],$$

*for sufficiently large fixed constants $c, d \geq 1$.*

*Proof.* LAPRAS runs the matrix mechanism on all queries in $P$ while the offline matrix mechanism runs on all queries in $\mathcal{S}$. It follows that

$$\sum_{q \in \mathcal{S}} \mathbb{E}[U_{LAPRAS}(q)^2] = \sum_{q \in P} \mathbb{E}[U_{LAPRAS}(q)^2]$$
$$+ \sum_{q \in \mathcal{S} \backslash P} \mathbb{E}[U_{LAPRAS}(q)^2],$$

by definition of our algorithm. Since $P \subseteq \mathcal{S}$, the optimization problem solved in the offline matrix mechanism contains all of the constraints solved by the matrix mechanism on $P$; thus, the error of the strategy matrix returned by LAPRAS is upper bounded by the strategy matrix returned by the offline matrix mechanism, and we have that $\sum_{q \in P} \mathbb{E}[U_{MM}(q)^2] \leq \sum_{q \in \mathcal{S}} \mathbb{E}[U_{MM}(q)^2]$. Hence, we can conclude that $\sum_{q \in P} \mathbb{E}[U_{LAPRAS}(q)^2] \leq c \sum_{q \in P} \mathbb{E}[U_{MM}(q)^2] \leq c \sum_{q \in \mathcal{S}} \mathbb{E}[U_{MM}(q)^2]$ for some sufficiently large fixed constant $c > 0$ where $c$ comes from our division of $\varepsilon$ in LAPRAS.

What remains is the bound on $\sum_{q \in \mathcal{S} \backslash P} \mathbb{E}[U^2_{LAPRAS}]$. We can upper bound this quantity by the sum of the variances of the independent noises for each of the $B$ bad queries drawn from the appropriate Gaussian distribution. Hence, $\sum_{q \in \mathcal{S} \backslash P} \mathbb{E}[U^2_{LAPRAS}] \leq B \cdot \sigma^2$ where $\sigma^2$ is the variance of the Gaussian distribution we are drawing noises from.

By Theorem 3.3 and Algorithm 1, the $\sigma$ of the distribution we use for the noise added to the bad queries is $\frac{c_1 \cdot B \cdot \sqrt{\ln(1/\delta)}}{\varepsilon}$ for some fixed constant $c_1 > 0$. Hence, we conclude that

$$\sum_{q \in \mathcal{S} \backslash P} \mathbb{E}[U^2_{LAPRAS}] \leq B \cdot \sigma^2$$
$$\leq \frac{c_1 B^3 \ln(1/\delta)}{\varepsilon^2}.$$

Finally, since the offline naive algorithm draws independent noises from the Gaussian distribution with $\sigma = \frac{c_2 M \sqrt{\ln(1/\delta)}}{\varepsilon}$ for a fixed constant $c_2 > 0$, the expected squared error of the queries answered by both the matrix mechanism and using Gaussian noise are upper bounded by the error given by drawing noise from this Gaussian distribution. Hence, for a sufficiently large fixed constant $d > 0$, it holds that

$$\frac{cB^3 \ln(1/\delta)}{\varepsilon^2} + c \sum_{q \in \mathcal{S}} \mathbb{E}[U_{MM}(q)^2] \leq d \cdot \sum_{q \in \mathcal{S}} \mathbb{E}[U_O(q)^2].$$

$\square$

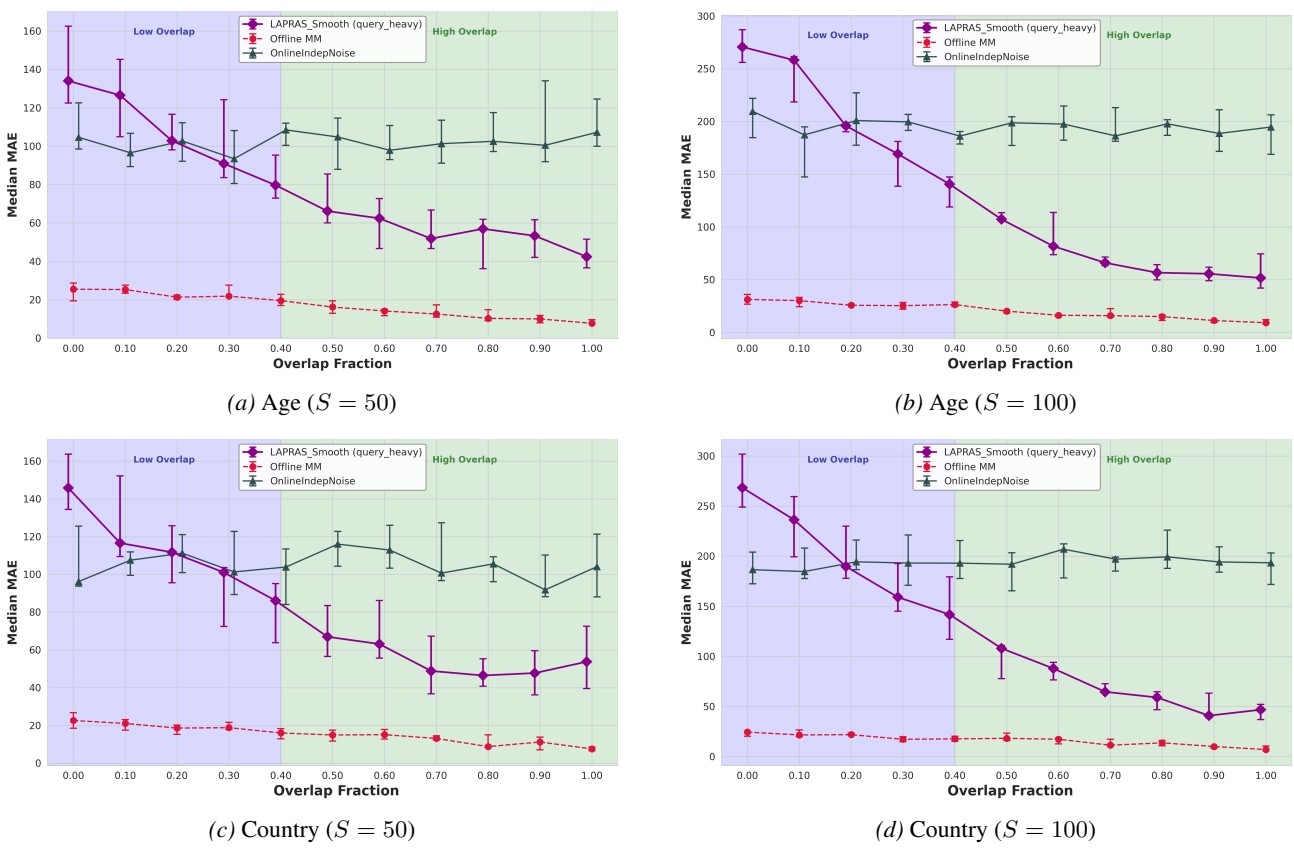

*(a)* Age $(S = 50)$

*(b)* Age $(S = 100)$

*(c)* Country $(S = 50)$

*(d)* Country $(S = 100)$

*Figure 1.* **ADULT dataset.** Median MAE (min, max bars) at $\varepsilon = 1.0$ for two attributes (Age, Country) for stream sizes $S \in \{50, 100\}$. [cite: 1358]

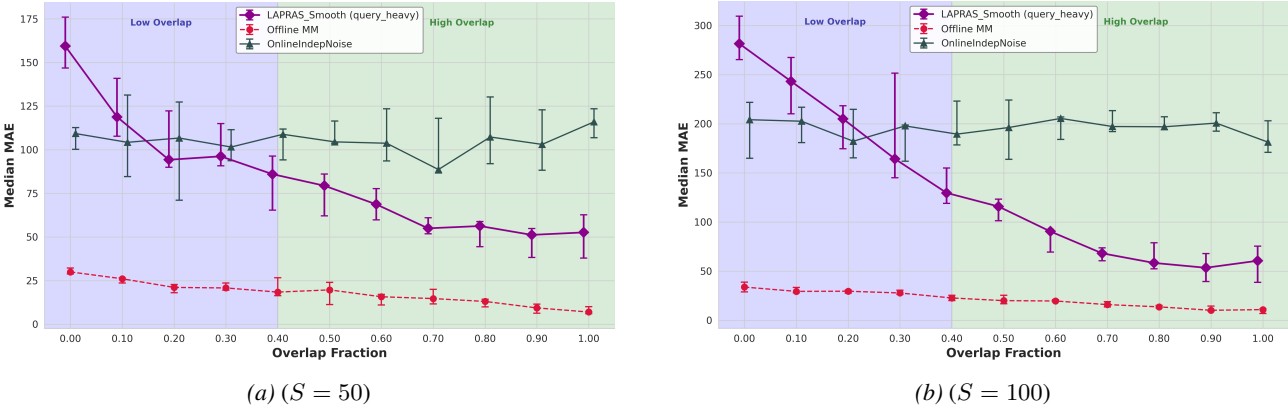

*(a)* $(S = 50)$

*(b)* $(S = 100)$

*Figure 2.* **Gowalla dataset.** Median MAE (min, max bar) at $\varepsilon = 1.0$ for stream sizes $S \in \{50, 100\}$.

## 5. Experimental Evaluation

We evaluate **LAPRAS** against standard baselines to test *consistency* (approaching offline utility when predictions are accurate) and *robustness* (remaining comparable to online mechanisms when predictions fail). For reproducibility, we generate workloads as described in Appendix B. We compare against **Online Independent Noise** (independent noise per query) and **OfflineMM** (matrix mech-

anism optimized for Q). To benchmark against existing reactive caching methods, we additionally compare against **CacheDP** ((Mazmudar et al., 2022)) under a strictly controlled, fixed global privacy budget. All experiments use $\varepsilon = 1.0$ and $\delta = 10^{-3}$, report two internal budget splits (*matrix_heavy*, *query_heavy*), and average over 5 runs. Our

*Table 2.* Median MAE versus overlap for **Adult** and **Gowalla** ($S = 100$, $\varepsilon = 1.0$).

| | ADULT | | | Gowalla | | |
|---|---|---|---|---|---|---|
| OVERLAP % | LAPRAS_Static (query_heavy) | LAPRAS_Static (matrix_heavy) | ONLINE INDEP. NOISE | LAPRAS_Static (query_heavy) | LAPRAS_Static (matrix_heavy) | ONLINE INDEP. NOISE |
| 0.0 | 201.7689 | 368.5622 | **186.5037** | 213.8990 | 391.1077 | **204.0514** |
| 0.1 | **181.1706** | 324.4794 | 184.7038 | **178.8394** | 317.4242 | 202.5222 |
| 0.2 | **145.9802** | 254.2288 | 194.3937 | **135.2609** | 234.7375 | 182.1618 |
| 0.3 | **119.6441** | 200.7279 | 193.1964 | **114.4690** | 184.5511 | 197.7361 |
| 0.4 | **105.7823** | 149.6603 | 193.0435 | **103.6537** | 156.0704 | 189.4034 |
| 0.5 | **89.0490** | 123.4891 | 191.9823 | **80.8661** | 114.8440 | 196.1295 |
| 0.6 | **66.7834** | 83.3435 | 207.0032 | **70.9335** | 85.4525 | 205.3693 |
| 0.7 | **62.2120** | 73.0511 | 197.0310 | **69.8578** | 79.4040 | 197.1238 |
| 0.8 | 80.7648 | **77.1926** | 199.3274 | 71.9180 | **64.7475** | 196.8623 |
| 0.9 | 62.2076 | **39.4620** | 194.2745 | 60.6568 | **42.6085** | 200.5918 |
| 1.0 | 43.0020 | **14.3340** | 193.4255 | 51.2221 | **17.0740** | 181.2074 |

code is available here[1].

**Datasets and Compute.** We use the Adult dataset (Kohavi, 1996), constructing a univariate histogram ($x \in \mathbb{R}^n$) for attributes: *age* and *country*. We also use the Gowalla check-in dataset (Cho et al., 2011), forming a histogram over locations and randomly subsampling ($N = 100$) locations. To test LAPRAS under a highly realistic, interactive access pattern, we additionally evaluate on a workload derived from IDEBench (Eichmann et al., 2020). Using the flights dataset and the ORIGIN_STATE_ABR family, we reduced IDE-style aggregate visualization queries into base linear counting queries, fully preserving the original filter predicates and exploratory query order. Experiments run on an Intel Xeon W-2145 (8 cores, 16 threads, up to 4.5GHz) with 64GB RAM[cite: 1368]. For the matrix-mechanism SDP, we use cvxpy (Diamond & Boyd, 2016; Agrawal et al., 2018) with the sdpa-python solver (Yamashita et al., 2003; 2012; Nakata, 2010; Kim et al., 2011).

**Overall Utility.** The overall performance of LAPRAS, as illustrated in Figure 1 and Figure 2, demonstrates a monotonic reduction in Mean Absolute Error (MAE) as the overlap percent increases. This trend confirms the fundamental premise of our approach: by exploiting workload predictability, LAPRAS bridges the gap between the hardness of the online model and the efficiency of the offline model. In regimes of high predictability, LAPRAS's utility converges toward the theoretical lower bound established by the Offline Matrix Mechanism. Conversely, as predictability degrades, the error profile smoothly transitions to match the Online Independent baseline. This behavior validates that our budget allocation strategies effectively hybridize the two paradigms, allowing the system to capitalize on *good* queries without suffering catastrophic failure on *bad* ones. Evaluation on the IDEBench dataset strongly confirms these synthetic trends: at 100% overlap, LAPRAS naturally

converges to near-perfect identical accuracy (MAE of 11.5).

**Static vs. Smooth Allocation.** While Static Allocation provides reliable worst-case guarantees, Smooth Allocation progressively overtakes it as prediction overlap improves. In head-to-head empirical evaluations keeping all parameters identical ($\varepsilon$, $\delta$, dataset, $|P|$, $|S|$), Smooth Allocation perfectly matches Static at $1.0$ overlap (by construction) and begins to yield significantly lower MAE once overlap exceeds 0.5 (Appendix B.4). In high-overlap scenarios ($\geq 0.7$), Smooth Allocation achieves up to $40\%$ lower MAE than Static, exhibiting consistent improvements across multiple normalized and absolute metrics (MAE, RMSE, NMAE, SMAPE). At lower overlaps ($< 0.5$), Static Allocation remains generally safer, as its uniform allocation strategy avoids overcommitting residual budget to poorly predicted future streams.

**Guidance on Budget Splits.** Empirically, the correct budget allocation strategy is dictated by expected prediction quality. When predictions are poor (overlap $\leq 0.5$), the *query_heavy* split achieves the lowest MAE by preserving more privacy budget for the online unpredicted queries. By contrast, when predictions are highly accurate (overlap $\geq 0.8$), *matrix_heavy* decisively outperforms other splits. Investing a larger fraction of the budget into the offline workload-aware matrix mechanism guarantees highly accurate precomputed answers that can be reused continuously across the stream.

**Robustness in Low Overlap Regime & Adversarial Arrivals.** We analyze the robustness of LAPRAS in the adversarial setting where the oracle fails to predict the workload. In this regime, the mechanism relies almost entirely on the allocation strategy to manage the online budget for the unpredicted stream. Table 2 reports the *Static Allocation* results: on Adult, the *query_heavy* split attains MAE 201.8 versus 186.5 for Online Independent Noise, and on Gowalla 213.9 versus 204.1, indicating only a small overhead from reserving budget for a prediction set that does not material-

---

[1] https://github.com/mundrapranay/learning-augmented-privacy/tree/icml2026

ize. Crucially, the error does not scale uncontrollably; the unbiased stopping-time estimator $\hat{B}$ successfully stabilizes noise levels even when the stream is dominated by unpredicted queries, confirming that LAPRAS retains worst-case online guarantees.

To further evaluate robustness against the random-order assumption, we tested a worst-case *bad-first* ordering, wherein all unpredicted queries arrive strictly before predicted queries (Appendix B.2). Under this severe adversarial pattern, Smooth Allocation degrades gracefully rather than failing catastrophically, showing at worst a $\approx 1.6\times$ MAE increase relative to Static Allocation. The *matrix_heavy* configuration proved exceptionally resilient to adversarial arrivals, effectively offering a stable accuracy floor due to its heavy reliance on the initial batch release.

**Sensitivity to Prediction Set Size and False Positives.** We explicitly evaluated the impact of large predicted sets and the resulting false positive rate (queries predicted but never realized). Varying $|P| \in \{50, 100, 200, 500\}$, we found that prediction overlap is the dominant factor for downstream utility, whereas $|P|$ and false positives are strictly secondary (Appendix B.1). For example, at an overlap of 1.0 with $|P| = 100$ and $S = 50$ (implying 50 false positives and a $50\%$ false positive rate), Smooth MAE remains extremely close (72.4) to the MAE seen with $|P| = 50$ (74.6). Thus, over-predicting does not materially hurt the mechanism's online utility. The primary constraint of a very large $P$ is the offline precomputation phase, as the SDP runtime scales roughly $\mathcal{O}(|P|^{1.5})$.

**Extending with a Query Cache (Smooth+Cache).** To optimize budget utilization on bad queries, we extended LAPRAS with a post-processing query cache (**Smooth+Cache**). All queries answered by the matrix mechanism are added to the cache. Upon receiving an unpredicted query, LAPRAS checks if it can be represented as a linear combination of cached queries via least-squares decomposition. If successful, the query is answered at zero additional privacy cost; if not, fresh budget is spent and the cache is updated. This addition yields consistent utility gains, particularly at moderate-to-high overlaps (0.3–0.8), where it leverages the robust initial matrix mechanism release to reduce MAE by up to $34.8\%$ relative to standard Smooth Allocation (Appendix B.5).

**Comparison with CacheDP.** We conducted a controlled comparison against CacheDP (Mazmudar et al., 2022) using identical global privacy budgets. Evaluated across scale-free normalized metrics (NMAE, NRMSE) on the Adult dataset, LAPRAS fundamentally outperforms CacheDP. When warming the cache with the predicted set queries prior to stream arrival, LAPRAS Static recorded an NMAE of 0.0019 at 0.0 overlap, compared to 0.2076 for CacheDP, a roughly $100\times$ difference. This disparity occurs because caching-based baselines distribute a single global budget linearly across queries, incurring large independent noise profiles, whereas LAPRAS pre-optimizes noise globally via the matrix mechanism (Appendix B.3).

**Optimality in High Overlap Regime.** In the high-overlap regime ($\rho \approx 1$), predictions cover most of the stream, and LAPRAS shifts effort from online noise to offline correlation-aware preprocessing. Table 2 shows that even the *query_heavy* split improves substantially over Independent Noise, but the gains are largest under *matrix_heavy*, as expected: allocating more budget to the batch component directly reduces the error of answers served from the precomputed state. Concretely, on Adult at $\rho = 1.0$, *matrix_heavy* attains MAE 14.3 versus 193.4 for Independent Noise, approaching the OfflineMM reference (6.96); on Gowalla, the corresponding values are 17.1 versus 181.2, again close to OfflineMM (10.8). These results confirm the intended behavior in the learning-augmented regime: when overlap is high, most queries are answered via the offline release with zero marginal privacy cost online, preserving the residual budget to cover the small unpredicted remainder and yielding near-offline utility in a streaming setting. Evaluated on the realistic IDEBench workload, LAPRAS Smooth similarly demonstrates extreme efficiency: at a moderate $50\%$ overlap with *matrix_heavy*, Smooth isolates the unpredicted queries perfectly, returning a 50.4 MAE compared to LAPRAS Static's 129.7, an impressive $61\%$ reduction in total error.

## 6. Conclusion

We introduced LAPRAS, a learning-augmented framework for online differentially private linear query answering that exploits workload predictability to narrow the online offline utility gap. Empirically, we show that predictions materially improve utility: LAPRAS approaches near optimal offline performance when overlap is high, yet remains comparable to the standard online baseline when predictions fail. Future work includes extending the estimator beyond random order to partially or fully adversarial arrivals, and replacing the offline component (e.g, HDMM (McKenna et al., 2020)).

## Acknowledgements

Pranay Mundra and Quanquan C. Liu were supported in part by the National Science Foundation (NSF) under Grant #CCF-2453323 and a Google Academic Research Award.

## Impact Statement

There are many potential societal consequences of our work including better accuracy guarantees for database systems under privacy constraints in practice.

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

## A. Theoretical Analysis

**Lemma A.1.** *Let a stream of size $S$ contain $B \geq T$ bad queries and $G = S - B$ good queries. Let $L$ be the random variable for the number of queries observed until the $T$-th bad query is seen. Let $T \geq c$ where $c$ is some small constant. The number of good queries seen, $Y = L - T$, follows a Negative Hypergeometric distribution, $Y \sim NHG(S, G, T)$*

*For any integer $k$ such that $1 \leq k < T$, the following identity for the inverse factorial moments of $L - 1$ holds:*

$$\mathbb{E}\left[\frac{(T-1)_k}{(L-1)_k}\right] = \frac{(B)_k}{(S)_k}$$

*where $(x)_k = x(x-1)\cdots(x-k+1)$ denotes the falling factorial.*

*Proof.* We proceed by induction on $k$.

**Base Case (k=1):** We must show that $\mathbb{E}\left[\frac{T-1}{L-1}\right] = \frac{B}{S}$. Let $D = \frac{T-1}{L-1} = \frac{T-1}{Y+T-1}$. The expectation is given by summing over the probability mass function (p.m.f.) of $Y \sim NHG(S, G, T)$.

$$\mathbb{E}\left[\frac{T-1}{L-1}\right] = \sum_{g=0}^{G} \frac{T-1}{g+T-1} \cdot P(Y = g)$$

$$= \sum_{g=0}^{G} \frac{T-1}{g+T-1} \cdot \frac{\binom{g+T-1}{g}\binom{S-T-g}{G-g}}{\binom{S}{G}}$$

$$= \frac{1}{\binom{S}{G}} \sum_{g=0}^{G} \frac{T-1}{g+T-1} \cdot \frac{(g+T-1)!}{g!(T-1)!}\binom{S-T-g}{G-g}$$

$$= \frac{1}{\binom{S}{G}} \sum_{g=0}^{G} \frac{(g+T-2)!}{g!(T-2)!}\binom{S-T-g}{G-g}$$

$$= \frac{1}{\binom{S}{G}} \sum_{g=0}^{G} \binom{g+T-2}{g}\binom{S-T-g}{G-g}$$

We use the Chu-Vandermonde Identity, which states: $\sum_{i=0}^{u} \binom{p+i}{i}\binom{q-i}{u-i} = \binom{p+q+1}{u}$ Let $p = T - 2$, $i = g$, $q = S - T$, and $u = G$. The sum becomes $\binom{(T-2)+(S-T)+1}{G} = \binom{S-1}{G}$.

$$\mathbb{E}\left[\frac{T-1}{L-1}\right] = \frac{1}{\binom{S}{G}}\binom{S-1}{G}$$

$$= \frac{G!(S-G)!}{S!} \cdot \frac{(S-1)!}{G!(S-1-G)!}$$

$$= \frac{(S-G)!}{S!} \cdot \frac{(S-1)!}{(S-G-1)!}$$

$$= \frac{S-G}{S} = \frac{B}{S}$$

Since $(B)_1 = B$ and $(S)_1 = S$, the base case holds.

**Inductive Step:** Assume for some integer $k \geq 1$ that the identity holds (Inductive Hypothesis):

$$\mathbb{E}\left[\frac{(T-1)_k}{(L-1)_k}\right] = \frac{(B)_k}{(S)_k}$$

We want to prove that it holds for $k + 1$:

$$\mathbb{E}\left[\frac{(T-1)_{(k+1)}}{(L-1)_{(k+1)}}\right] = \frac{(B)_{(k+1)}}{(S)_{(k+1)}}$$

Let's evaluate the expectation on the left-hand side.

$$\mathbb{E}\left[\frac{(T-1)_{(k+1)}}{(L-1)_{(k+1)}}\right] = \sum_{g=0}^{G} \frac{(T-1)_{k+1}}{(g+T-1)_{k+1}} \cdot \frac{\binom{g+T-1}{g}\binom{S-T-g}{G-g}}{\binom{S}{G}}$$

Consider the term involving the falling factorials and the first binomial coefficient:

$$\frac{(T-1)_{k+1}}{(g+T-1)_{k+1}}\binom{g+T-1}{g} = \frac{(T-1)!/(T-k-2)!}{(g+T-1)!/(g+T-k-2)!} \cdot \frac{(g+T-1)!}{g!(T-1)!}$$

$$= \frac{(T-1)!}{(T-k-2)!} \cdot \frac{(g+T-k-2)!}{(g+T-1)!} \cdot \frac{(g+T-1)!}{g!(T-1)!}$$

$$= \frac{(g+T-k-2)!}{g!(T-k-2)!} = \binom{g+T-k-2}{g}$$

Substituting this back into the expectation sum:

$$\mathbb{E}\left[\frac{(T-1)_{(k+1)}}{(L-1)_{(k+1)}}\right] = \frac{1}{\binom{S}{G}} \sum_{g=0}^{G} \binom{g+T-k-2}{g}\binom{S-T-g}{G-g}$$

Again, we apply the Chu-Vandermonde Identity with $p = T-k-2$, $i = g$, $q = S-T$, and $u = G$. The sum becomes $\binom{(T-k-2)+(S-T)+1}{G} = \binom{S-k-1}{G}$.

$$\mathbb{E}\left[\frac{(T-1)_{(k+1)}}{(L-1)_{(k+1)}}\right] = \frac{1}{\binom{S}{G}}\binom{S-k-1}{G}$$

$$= \frac{G!(S-G)!}{S!} \cdot \frac{(S-k-1)!}{G!(S-k-1-G)!}$$

$$= \frac{(S-G)!}{S!} \cdot \frac{(S-k-1)!}{(S-G-k-1)!}$$

$$= \frac{B!}{S!} \cdot \frac{(S-k-1)!}{(B-k-1)!}$$

$$= \frac{B!/(B-(k+1))!}{S!/(S-(k+1))!} = \frac{(B)_{(k+1)}}{(S)_{(k+1)}}$$

This completes the inductive step.

By the principle of mathematical induction, the identity holds for all integers $k \geq 1$. $\qquad\square$

**Theorem 4.3.** *For a fixed integer $T \geq 3$, the variance of the estimator $\hat{B}$ is bounded by:*

$$\mathrm{Var}[\hat{B}] < \frac{S(T-1)}{(S-1)(T-2)}B(B-1) - B^2$$

*where $B$ is the number of bad queries and $S$ is the total number of queries.*

*Proof.* Recall that $\hat{B} = S \cdot D$, where $D = \frac{T-1}{L-1}$, and $L = Y + T$ is the number of queries until the $T$-th bad query is observed. The variance is given by $\mathrm{Var}[\hat{B}] = S^2\mathrm{Var}[D] = S^2(\mathbb{E}[D^2] - (\mathbb{E}[D])^2)$.

Our goal is to find a tighter upper bound for $\mathbb{E}[D^2]$. Using Lemma A.1 (for $k$=2), we have:

$$\mathbb{E}\left[\frac{(T-1)_2}{(L-1)_2}\right] = \mathbb{E}\left[\frac{(T-1)(T-2)}{(L-1)(L-2)}\right] = \frac{B(B-1)}{S(S-1)}$$

This identity holds for $T \geq 3$, which ensures that $L-2$ is always positive, as $L \geq T$. Consider the term $\frac{1}{(L-1)^2}$. For any $L \geq 3$, we can establish the following strict inequality:

$$L-2 < L-1 \implies \frac{1}{(L-1)(L-2)} > \frac{1}{(L-1)^2}$$

This inequality allows us to bound the expectation. By taking the expectation of both sides (which preserves the inequality for non-constant random variables), we get:

$$\mathbb{E}\left[\frac{1}{(L-1)^2}\right] < \mathbb{E}\left[\frac{1}{(L-1)(L-2)}\right]$$

Now, we can use this to bound $\mathbb{E}[D^2]$:

$$\mathbb{E}[D^2] = \mathbb{E}\left[\left(\frac{T-1}{L-1}\right)^2\right]$$

$$= (T-1)^2\,\mathbb{E}\left[\frac{1}{(L-1)^2}\right]$$

$$< (T-1)^2\,\mathbb{E}\left[\frac{1}{(L-1)(L-2)}\right]$$

We can rewrite the expectation term to relate it to our known identity:

$$\mathbb{E}\left[\frac{1}{(L-1)(L-2)}\right] = \frac{1}{(T-1)(T-2)}\,\mathbb{E}\left[\frac{(T-1)(T-2)}{(L-1)(L-2)}\right]$$

$$= \frac{1}{(T-1)(T-2)} \cdot \frac{B(B-1)}{S(S-1)}$$

Substituting this back into our inequality for $\mathbb{E}[D^2]$:

$$\mathbb{E}[D^2] < (T-1)^2 \left(\frac{1}{(T-1)(T-2)}\frac{B(B-1)}{S(S-1)}\right)$$

$$= \frac{T-1}{T-2}\frac{B(B-1)}{S(S-1)}$$

Now we substitute this tighter bound into the variance formula for $D$:

$$\mathrm{Var}[D] = \mathbb{E}[D^2] - (\mathbb{E}[D])^2 < \frac{T-1}{T-2}\frac{B(B-1)}{S(S-1)} - \left(\frac{B}{S}\right)^2$$

Finally, we find the variance of $\hat{B} = S \cdot D$:

$$\mathrm{Var}[\hat{B}] = S^2\mathrm{Var}[D] < S^2\left(\frac{T-1}{T-2}\frac{B(B-1)}{S(S-1)} - \frac{B^2}{S^2}\right)$$

$$= \frac{S^2(T-1)B(B-1)}{S(S-1)(T-2)} - \frac{S^2B^2}{S^2}$$

$$= \frac{S(T-1)}{(S-1)(T-2)}B(B-1) - B^2$$

This concludes the proof for the sharper upper bound. $\qquad\square$

**Corollary 4.4.** *With $T = \lceil\log^2(S)\rceil$, the variance of the estimator $\hat{B}$ has the following asymptotic upper bound:*

$$\mathrm{Var}[\hat{B}] = O\left(\frac{B^2}{\log^2(S)}\right)$$

*Proof.* We begin with the bound from Theorem 4.3. Let's first rearrange the expression:

$$\mathrm{Var}[\hat{B}] < \frac{S(T-1)}{(S-1)(T-2)}B(B-1) - B^2$$

$$= \left(\frac{S(T-1)}{(S-1)(T-2)} - 1\right)B^2 - \frac{S(T-1)}{(S-1)(T-2)}B$$

Our goal is to analyze the coefficients as $S \to \infty$. Let's analyze the first coefficient:

$$\frac{S(T-1)}{(S-1)(T-2)} - 1 = \frac{S(T-1) - (S-1)(T-2)}{(S-1)(T-2)}$$
$$= \frac{(ST-S) - (ST-2S-T+2)}{(S-1)(T-2)}$$
$$= \frac{S+T-2}{(S-1)(T-2)}$$

Now, we substitute $T = \log^2(S)$. For large $S$, the dominant term in the numerator is $S$ and in the denominator is $S \cdot T = S \log^2(S)$. The asymptotic behavior is:

$$\frac{S+T-2}{(S-1)(T-2)} \sim \frac{S}{S \log^2(S)} = \frac{1}{\log^2(S)}$$

For the second coefficient, as $S \to \infty$, we have $\frac{S}{S-1} \to 1$ and $\frac{T-1}{T-2} = \frac{\log^2(S)-1}{\log^2(S)-2} \to 1$. Thus, the entire coefficient $\frac{S(T-1)}{(S-1)(T-2)} \to 1$.

Combining these results, the variance bound behaves as:

$$\text{Var}[\hat{B}] \approx \frac{1}{\log^2(S)} B^2 - B$$

The dominant term in this expression is the one involving $B^2$.

$$\text{Var}[\hat{B}] = O\left(\frac{B^2}{\log^2(S)}\right)$$

$\square$

**Lemma 4.5.** *Let $\varepsilon_{\text{pool}}$ be the initial budget allocated to the Smooth Allocation strategy and $B$ be the total count of bad queries. The total budget expended satisfies:*

$$\sum_{i=2}^{B} \varepsilon_i < \varepsilon_{\text{pool}} \tag{1}$$

*Proof.* Let $\varepsilon_{\text{rem},i}$ denote the budget remaining after answering the $i$-th bad query, with $\varepsilon_{\text{rem},1} = \varepsilon_{\text{pool}}$. The allocation rule is defined as $\varepsilon_i = \varepsilon_{\text{rem},i-1}/(\hat{B}_{\text{rem},i} + 1)$, where $\hat{B}_{\text{rem},i} \geq 1$.

The recurrence relation for the remaining budget is:

$$\varepsilon_{\text{rem},i} = \varepsilon_{\text{rem},i-1} - \varepsilon_i$$
$$= \varepsilon_{\text{rem},i-1}\left(1 - \frac{1}{\hat{B}_{\text{rem},i} + 1}\right)$$
$$= \varepsilon_{\text{rem},i-1}\left(\frac{\hat{B}_{\text{rem},i}}{\hat{B}_{\text{rem},i} + 1}\right).$$

Let $\alpha_i = \frac{\hat{B}_{\text{rem},i}}{\hat{B}_{\text{rem},i}+1}$. Since $\hat{B}_{\text{rem},i} \geq 1$, it holds that $0 < \alpha_i < 1$ for all $i$. The final remaining budget after $B$ queries is obtained by unrolling the recurrence:

$$\varepsilon_{\text{rem},B} = \varepsilon_{\text{pool}} \prod_{k=2}^{B} \alpha_k. \tag{2}$$

Since $\varepsilon_{\text{pool}} > 0$ and $\alpha_k > 0$ for all $k$, the product is strictly positive, implying $\varepsilon_{\text{rem},B} > 0$. The total spent budget is $E_{\text{spent}} = \varepsilon_{\text{pool}} - \varepsilon_{\text{rem},B}$. Given $\varepsilon_{\text{rem},B} > 0$, it follows that $E_{\text{spent}} < \varepsilon_{\text{pool}}$. $\square$

# B. Experimental Evaluation

In this section, we provide extended experimental results that complement the main paper, including sensitivity analysis on the predicted set size, robustness under adversarial arrival orders, normalized error metrics, comparisons with caching-based baselines, and an ablation of our allocation strategies.

**Query Families** Let the domain have size n. We construct two families: - Range queries $U_{rng}$: all contiguous half-open intervals $[i : j]$ with $i < j$, represented as $q \in \{0, 1\}^n$ with ones on indices $i, \ldots, j - 1$ and zeros elsewhere. - Random binary queries $U_{rand}$: a fixed-size collection of random linear queries $q \in \mathbb{R}^n$.

**Query Stream** The oracle is provided the universe $U_{rng}$ and returns a predicted set $P$ by uniform sampling without replacement. The query stream $S$ is formed by taking $\lfloor \rho S \rfloor$ queries sampled without replacement from $P$ and $(S - \lfloor \rho S \rfloor)$ queries sampled without replacement from $U_{rand}$, followed by a random permutation.

## B.1. Sensitivity to Predicted Set Size and False Positives

*Table 3.* Sensitivity of LAPRAS to Predicted Set Size ($|P|$) and False Positive Rate (FPR). Median MAE is reported for the Adult dataset using *query_heavy* allocation ($S = 100, Q = 50, \varepsilon = 1.0$).

| $|P|$ | Overlap | FPR | Static MAE | Smooth MAE | Offline MM | Offline IN |
|---|---|---|---|---|---|---|
| 50 | 0.0 | 100% | 216.0 | 304.3 | 39.2 | 207.4 |
| 50 | 0.5 | 50% | 125.0 | 116.3 | 26.7 | 206.6 |
| 50 | 1.0 | 0% | 74.6 | 74.6 | 10.8 | 206.7 |
| 100 | 0.0 | 100% | 194.6 | 325.7 | 35.4 | 202.8 |
| 100 | 0.5 | 75% | 97.1 | 107.0 | 24.6 | 203.6 |
| 100 | 1.0 | 50% | 72.4 | 72.4 | 13.8 | 192.3 |

*Table 4.* Runtime Scaling with Predicted Set Size ($|P|$). Runtimes reflect the median across overlaps for the *query_heavy* strategy.

| $|P|$ | $Q$ | LAPRAS MM (s) | LAPRAS Smooth (s) | Offline MM (s) |
|---|---|---|---|---|
| 50 | 50 | 185.8 | 0.008 | 170.6 |
| 100 | 50 | 231.7 | 0.008 | 142.1 |
| 200 | 500 | 356.9 | 0.100 | 596.6 |
| 500 | 500 | 874.6 | 0.200 | 627.3 |

## B.2. Robustness to Adversarial Arrival Order

To test the limits of our random-order assumption, we evaluated LAPRAS under a strict *bad-first* adversarial ordering, where all unpredicted queries arrive before any predicted queries.

*Table 5.* LAPRAS MAE under Adversarial Arrival Order (*bad-first*). Settings: Gowalla, $|P| = 100, Q = 100, S = 100, \varepsilon = 1.0$.

| | Query Heavy | | Matrix Heavy | | | |
|---|---|---|---|---|---|---|
| Overlap | Static | Smooth | Static | Smooth | Offline MM | Offline IN |
| 0.0 | 209.5 | 257.4 | 382.8 | 334.4 | 34.7 | 196.0 |
| 0.3 | 169.2 | 209.3 | 277.2 | 248.2 | 28.6 | 208.6 |
| 0.5 | 130.8 | 162.4 | 205.0 | 186.7 | 21.9 | 203.8 |
| 0.7 | 83.5 | 121.2 | 98.5 | 120.9 | 17.3 | 193.8 |
| 1.0 | 44.5 | 44.5 | 14.8 | 14.8 | 8.1 | 183.6 |

## B.3. Normalized Error Metrics and CacheDP Comparison

To ease interpretation across varying count scales, we report scale-free normalized metrics (NMAE, NRMSE, MAPE, SMAPE). Furthermore, we conducted a controlled comparison against CacheDP under a fixed global privacy budget

constraint.

*Table 6.* Scale-free Error Metrics for LAPRAS Smooth (*query_heavy*, $|P| = 100$, $Q = 50$). Values are scaled by $\times 100$ for readability.

| $|P|$ | Overlap | NMAE | NRMSE (Range) | MAPE | SMAPE |
|---|---|---|---|---|---|
| 50 | 0% | 114.0 | 141.1 | 1946.6 | 1.6 |
| 50 | 50% | 35.5 | 50.6 | 809.5 | 1.2 |
| 50 | 100% | 31.5 | 38.1 | 164.2 | 0.9 |
| 100 | 0% | 140.6 | 175.5 | 2383.6 | 1.7 |
| 100 | 50% | 32.3 | 46.5 | 603.9 | 1.2 |
| 100 | 100% | 21.0 | 26.7 | 189.6 | 0.8 |

*Table 7.* Comparison of LAPRAS against CacheDP on the Adult dataset ($|P| = 100$, $Q = 100$, $S = 100$, $\varepsilon = 1.0$). Metrics are normalized scale-free errors.

| | **NMAE** | | | **NRMSE (Range-Norm)** | | |
|---|---|---|---|---|---|---|
| Overlap | Static | Smooth | CacheDP | Static | Smooth | CacheDP |
| 0.0 | 0.0019 | 0.0026 | 0.2076 | 0.0025 | 0.0033 | 0.2569 |
| 0.5 | 0.0007 | 0.0011 | 0.1242 | 0.0010 | 0.0017 | 0.1944 |
| 1.0 | 0.0014 | 0.0014 | 0.0588 | 0.0018 | 0.0018 | 0.2011 |

## B.4. Ablation: Static vs. Smooth Allocation

We compare Static and Smooth allocations under identical stream conditions to identify the crossover point where prediction overlap yields utility gains for dynamic pacing.

*Table 8.* Direct MAE comparison between Static and Smooth Allocation ($|P| = 50$, $Q = 50$, $\varepsilon = 1.0$). Positive improvement indicates Smooth outperforms Static.

| | **Matrix Heavy** | | | **Query Heavy** | | |
|---|---|---|---|---|---|---|
| Overlap | Static | Smooth | % Improv. | Static | Smooth | % Improv. |
| 0.5 | 184.7 | 166.3 | +10.0% | 125.0 | 116.3 | +6.9% |
| 0.6 | 160.3 | 114.0 | +28.8% | 117.0 | 106.5 | +9.0% |
| 0.7 | 137.3 | 84.5 | +38.4% | 103.5 | 82.5 | +20.3% |
| 0.8 | 87.0 | 53.7 | +38.2% | 90.0 | 86.2 | +4.3% |
| 0.9 | 59.0 | 35.2 | +40.2% | 78.7 | 65.7 | +16.5% |
| 1.0 | 24.9 | 24.9 | 0.0% | 74.6 | 74.6 | 0.0% |

## B.5. Extending LAPRAS with a Query Cache (Smooth+Cache)

We evaluated an extension of our algorithm that maintains a cache of precomputed predicted queries. If an unpredicted query lies in the linear span of the cache, it is answered via post-processing at zero additional privacy cost.

*Table 9.* Effect of adding caching (Smooth+Cache) to LAPRAS ($|P| = 100$, $Q = 50$, $\varepsilon = 1.0$).

| Overlap | **Query Heavy** | | | **Matrix Heavy** | | |
|---------|--------|-------------|-----------------|--------|-------------|-----------------|
|         | Smooth | Smooth+Cache | Cache vs Smooth | Smooth | Smooth+Cache | Cache vs Smooth |
| 0.0 | 325.7 | 304.7 | +6.4% | 612.4 | 578.8 | +5.5% |
| 0.3 | 185.1 | 174.0 | +6.0% | 325.1 | 296.8 | +8.7% |
| 0.5 | 107.0 | 110.5 | -3.3% | 161.5 | 150.3 | +6.9% |
| 0.7 | 86.4 | 86.8 | -0.5% | 85.6 | 87.3 | -2.0% |
| 0.8 | 78.3 | 71.2 | +9.1% | 64.1 | 41.8 | +34.8% |
| 1.0 | 72.4 | 72.4 | 0.0% | 24.1 | 24.1 | 0.0% |

