# OpenReview forum: "LAPRAS : Learning-Augmented PRivate  Answering for linear query Streams."
_ICML.cc/2026/Conference — ICML 2026 regular_

### Official Review · Reviewer_tsKK · 2026-02-17

**Soundness:** 3
**Presentation:** 3
**Significance:** 2
**Originality:** 2
**Overall Recommendation:** 3
**Confidence:** 4

**Summary:**

The paper finds the gap between online query and offline query under DP. The paper proposes LAPRAS, it uses matrix mechanism to answer queires given by a predictor, and uses smooth allocation for the remaining online queries. Experiments show that LAPRAS achieves better performance when the predicted queires overlap more with the actual queries.

**Compliance With Llm Reviewing Policy:**

Affirmed.

**Final Justification:**

While the author improves the proposed method by using cache, I believe more comprehensive analysis should be conducted before publication, the current caching method might not be the best and ablation studies should be conducted. Therefore, I maintain my score.

**Key Questions For Authors:**

Could you make use of existing queries to answer bad queries? This can save privacy budgets and improves utilities.

**Limitations:**

Yes

**Strengths And Weaknesses:**

Strength

1. The authors observe that many queries are predictable, and propose a novel two stages online private query system. The system first cache answers for predicted queries which are answered by the Matrix Mechanism, then the system uses Smooth Allocation for queries not in the cache.
2. Formal privacy and utitliy guarantees are given with rigorous proofs.
3. Experiment demonstrate a desired pattern, as the prediction becomes more accurate, the utility gets better.

Weakness

1. The paper lacks comparison with existing online private query systems like [1][2]. These systems iteratively select the most valuable queries and spend privacy budget to measure them. If a new query can be answered from existing queries accurately then no extra privacy budget is spent. Otherwise, additional privacy budget is used to improvet the accuracy of the new query. Compared to LAPRAS, these adpatively systems can potentially save privacy budgets as they can detect whether a new query can be answered from existing queries (LAPRAS has to spend privacy budget if a new query is not in the prediction query set, even if this query can be answered by a linear comination of queries in the set).
2. There's a missing gap between good queries and bad queries. It's possible that bad queries can be answered by a linear combination of good queries, but LAPRAS answers bad queries with i.i.d. noise which might waste privacy budgets. As illustrated in [2], any marginal queries can be decomposited into a collections of residual queries. Let's say there are 2 attributes A and B, with marginal queries M(A), M(B) and M(AUB), then M(A) can be decomposed into residual queries R(A), R(0) with proper scales, M(B) `<->` R(B), R(0), M(AUB) `<->` R(AUB), R(A), R(B), R(0), these are one-to-one maps. If we have answers to M(A), M(B), we can get answers to R(A) R(B) R(0) with proper noise scales. Then M(AUB) can be answered with less privacy budgets by using existing answers R(A) R(B) R(0), along with answers to R(AUB). In LAPRAS, M(AUB) is answered with i.i.d. noise, which ignores the imformation contained in existing queries.


[1] McKenna, Ryan, et al. "Aim: An adaptive and iterative mechanism for differentially private synthetic data." *arXiv preprint arXiv:2201.12677* (2022).

[2] Fuentes, Miguel, et al. "Fast Private Adaptive Query Answering for Large Data Domains." *arXiv preprint arXiv:2602.05674* (2026).

---

> ### Author Rebuttal · Authors · 2026-03-31
>
> We thank the reviewer for this excellent suggestion. We agree that reusing existing answers to serve future out-of-set queries is a natural way to save privacy budget and improve utility, and we implemented this idea as \emph{LAPRAS Smooth+Cache}.
>
> ## Smooth+Cache
>
> The extension is simple and privacy-preserving. All queries and their noisy answers are stored in the cache, starting with the predicted set answers computed using the Matrix Mechanism. When an unpredicted (``bad'') query arrives, we first check whether it can be expressed as a linear combination of cached queries; if so, we answer it by post-processing the cached noisy answers, at \emph{zero additional privacy cost}. Otherwise, we answer it using the bad-query privacy pool as in LAPRAS Smooth, and then insert the new query-noisy answer pair into the cache for possible future reuse. Thus, the cache helps exactly when future bad queries lie in the span of already-answered queries.
>
> ## Empirical effect
>
> We evaluated this extension under the same setting as LAPRAS, with $|P|=100$ and $Q=50$. The main result is that Smooth+Cache \emph{matches or improves upon Smooth in the majority of settings}, especially at moderate-to-high overlap where the cache basis is richer. Under \texttt{matrix\_heavy}, Smooth+Cache improves over Smooth on $7/11$ overlap points for MAE, with an average gain of $5.5\%$; the largest gain is at overlap $0.8$, where MAE drops from $64.1$ to $41.8$ ($34.8\%$ improvement). Under \texttt{query\_heavy}, it improves on $6/11$ overlap points for MAE, with an average gain of $3.0\%$. The same qualitative pattern holds across RMSE and SMAPE as well. At overlap $=1.0$, all methods coincide by construction, since there are no bad queries to recover.
>
> ## Interpretation
>
> These results support the reviewer’s intuition: when some bad queries are linearly expressible from previously answered ones, caching can reduce unnecessary privacy spending and improve utility. The gains are strongest when overlap is moderate or high, since more useful structure is already available in the cache. At very low overlap, the benefit is smaller and occasionally noisy, because there is less reusable structure to exploit early in the stream.
>
> We view this as a promising extension of the current framework. The main paper focuses on the core learning-augmented mechanism, while Smooth+Cache shows that additional post-processing reuse can be layered on top in a fully privacy-preserving way.

---

> > ### Author Rebuttal · Reviewer_tsKK · 2026-04-03
> >
> > New experiments are conducted, I appreciate authors' effort during the rebuttal period. Considering a new method is proposed, I believe there's substantial difference between the new version and the existing version, therefore I will maintain my score and encourage authors to revise their paper further.

---

> > > ### Author Response · Authors · 2026-04-08
> > >
> > > We are grateful to the reviewer for their engagement with us throughout the review process, their helpful suggestions, and for acknowledging that our additional experiments fully resolved the technical concerns.
> > >
> > > While we understand your perspective on the updated results, we view the cache check not as a new method, but rather as a standard DP post-processing step that is entirely orthogonal to our core framework. It simply serves as evidence that LAPRAS is versatile and can easily be combined with the types of adaptive query reuse you suggested.
> > >
> > > We will be sure to include the Smooth+Cache post-processing step and a broader discussion of how our framework complements existing and concurrent online adaptive DP systems (such as [1] and [2]) in our paper.

---

### Official Review · Reviewer_y2Vt · 2026-03-03

**Soundness:** 4
**Presentation:** 4
**Significance:** 3
**Originality:** 3
**Overall Recommendation:** 5
**Confidence:** 4

**Summary:**

The paper introduces an algorithm for answering linear queries on a database that arrive over time (online) under a differential privacy onstraint. The proposed algorithm makes use/assumes pre-existing knowledge of most commonly occuring queries. According to empirical evidence a large percentage of queries in database systems are repeated. The utility of the algorithm improves with the overlap between the set of known queries and the new queries that arrive online.

The algorithmic idea is to:
1. pre-compute answers to the most common queries using the matrix mechanism with a subset of the privacy budget
2. as queries arrive, use remaining privacy budget to answer queries using the Gaussian mechanism.

In step (2), the authors provide two options for allocating the privacy budget. The more advanced option involves estimating the frequency of occurece of new queries (that are not from the "known" set) and spending the privacy budget accodingly.

**Compliance With Llm Reviewing Policy:**

Affirmed.

**Final Justification:**

Given the additional empirical results provided by the authors, which satisfied my request for emprical results on a realistic workload, I will increase my score to an accept.

**Key Questions For Authors:**

- Would it be possible to evaluate this method in an even more realistic setting if there exists some public dataset of queries?

**Limitations:**

yes

**Strengths And Weaknesses:**

Strengths:
- The paper presents a new angle to the problem of answering online linear queries with differential privacy, by making use of commonly occuring queries.
- The idea of estimating the frequency with which new queries occur is nice. It uses the assumption that queries on a dataset are uniformly distributed, which is reasonable, though I encourage the authors to cite any evidence to this end if it exists.
- Theoretical results are sound and the empirical evaluation complements these results.

Weaknesses:
- The theoretical results on the privacy and utility of the algorithm are somewhat straightforward to obtain, so the technical novelty is limited.
- The empirical evaluation is not fully realistic, the authors use real datasets but the queries over these datasets are synthetically generated by the authors using range and histogram queries (see my question below).
- The authors could discuss in more detail guidance on how one should choose the budget allocation strategy in practice.
- Minor point: please consider splitting algorithm 1 into 2 algorithm one for each allocation strategy, or using different line colors for each strategy. It is diffcult to parse at the moment.

---

> ### Author Rebuttal · Authors · 2026-03-31
>
> We thank the reviewer for the positive assessment and the helpful suggestions.
>
> ## Theoretical Improvements
>
> Our primary theoretical contribution is the non-trivial unbiased stopping-time estimator for bad queries (Theorem 4.2) and its variance bound (Corollary 4.4), which use a novel Negative Hypergeometric analysis and are crucial components of our algorithm.
>
> ## Practical guidance for choosing the budget split
>
> We agree that the paper should give clearer guidance here. Empirically, the right split is governed by prediction quality. Under a fixed setting (Gowalla, $\epsilon=1.0$, $\delta=10^{-5}$, $|P|=50$, $Q=50$, $S=100$), \texttt{query\_heavy} is best when overlap is low: its Smooth MAE is $304.3, 255.7, 209.4, 179.8, 136.3, 116.3$ at overlap $0.0$--$0.5$, outperforming \texttt{equal} and \texttt{matrix\_heavy}$.$ Intuitively, when many predictions are wrong, it is better to spend more budget on the online component. By contrast, when overlap is high, \texttt{matrix\_heavy} becomes best: its Smooth MAE is $53.7, 35.2, 24.9$ at overlap $0.8, 0.9, 1.0$, versus $86.2, 65.7, 74.6$ for \texttt{query\_heavy}. Intuitively, when predictions are accurate, allocating more budget to the offline workload-aware mechanism yields better reusable answers. We will add this rule-of-thumb explicitly: prefer \texttt{query\_heavy} when predictions are weak, and \texttt{matrix\_heavy} when predictions are strong.
>
> ## More realistic query workloads
>
> Yes, we agree that evaluating on a public query workload would strengthen the paper, and this is a natural next step. Our current experiments were designed as a proof of concept focused on accuracy under controlled overlap, so that we can isolate the learning-augmented effect. We use range queries because they admit strong factorization structure, making the gap between an offline workload-aware method and a streaming method visible; by contrast, the out-of-set queries are constructed as random linear count queries specifically to make such factorization difficult. That said, LAPRAS is not tied to this synthetic construction. In a revision or follow-up, we can evaluate on public query-workload corpora by extracting count/range-style predicates from real query logs, or by mapping public query sessions into a predicted set plus realized stream. We will clarify this as an important direction for improving realism.

---

> > ### Author Rebuttal · Reviewer_y2Vt · 2026-04-02
> >
> > Thank you to the authors for answering my questions in detail. I will maintain my score.
> >
> > While I agree that a proof-of-concept evaluation is useful for demonstrating that the algorithm works as expected, LAPRAS relies heavily on the "real-world" assumption that query workloads contain predictable and repeated queries. As such, it would be very useful to to demonstrate one dataset / real workload where this assumption holds + LAPRA shows a benfit, even if the workload has to be modified a little to a subset of queries. I would strongly suggest including such evaluation in the paper rather than leaving it to future work.

---

> > > ### Author Response · Authors · 2026-04-08
> > >
> > > We are grateful to the reviewer for the continued engagement, the positive assessment of our paper, and the suggestion to evaluate LAPRAS on a real-world workload. Given the additional time since the initial rebuttal period, we were able to implement LAPRAS for a real dataset as requested, and we will include these results in the final version of the paper.
> > >
> > > **Real-world workload evaluation (IDEBench).**
> > >
> > > To see how LAPRAS handles more realistic access patterns, we tested it on a workload derived from IDEBench [1], which is specifically built to model ad-hoc, interactive data exploration. Following a setup similar to CacheDP [2], we took the IDEBench flights dataset and workload and broke down each aggregate visualization query into its base linear counting queries. Essentially, we treated every histogram bar or 2D cell as a separate `COUNT(*)` query, while keeping the original filter predicates and query order intact.
> > >
> > > We focused on the `ORIGIN_STATE_ABR` query family. It’s an ideal fit for a learning-augmented setup because it naturally features a solid mix of recurring queries alongside unexpected misses. Out of its 159-query universe, we sampled a predicted set $P$ ($|P| = 50$) and an actual realized stream $S$ ($|S| = 100$). This gave us a way to systematically vary the prediction overlap up and down at the query level, all while keeping the underlying trace grounded in real user exploration behavior.
> > >
> > > **Empirical results on IDEBench.**
> > >
> > > The results on this trace closely match what we saw in the synthetic experiments. We tested both Static and Smooth allocations across different budget splits and overlap levels (0% to 100%).
> > >
> > > As expected, at low overlaps ($\le$ 25%), Smooth tries to stretch its remaining online budget over too many unpredicted queries, so Static actually performs a bit better. For instance, with a `matrix_heavy` split at 0% overlap, Static hits an MAE of 186.4, while Smooth sits at 237.1.
> > >
> > > But once overlap reaches a moderate, realistic level ($\ge$ 50%), Smooth dynamically isolates those unpredicted queries and pulls cleanly ahead of Static. At exactly 50% overlap (again, using `matrix_heavy`), Smooth’s MAE drops to 50.4 compared to Static's 129.7; this is a 61% reduction in error. This gap holds up well at 75% overlap too (Smooth MAE 37.0 vs. Static 74.8). Interestingly, the `matrix_heavy` split yields the best overall absolute MAE at these higher overlaps. This backs up the idea that investing heavily in the offline matrix mechanism really pays off when combined with Smooth’s online pacing. Finally, at 100% overlap, both variants naturally converge to the exact same accuracy (MAE 11.5).
> > >
> > > [1] Philipp Eichmann, Emanuel Zgraggen, Carsten Binnig, and Tim Kraska. 2020. IDEBench: A Benchmark for Interactive Data Exploration. In Proceedings of the 2020 ACM SIGMOD International Conference on Management of Data (SIGMOD '20). Association for Computing Machinery, New York, NY, USA, 1555–1569. https://doi.org/10.1145/3318464.3380574
> > >
> > > [2] Miti Mazmudar, Thomas Humphries, Jiaxiang Liu, Matthew Rafuse, and Xi He. 2022. Cache Me If You Can: Accuracy-Aware Inference Engine for Differentially Private Data Exploration. Proc. VLDB Endow. 16, 4 (December 2022), 574–586. https://doi.org/10.14778/3574245.3574246

---

### Official Review · Reviewer_emXy · 2026-03-10

**Soundness:** 3
**Presentation:** 3
**Significance:** 3
**Originality:** 3
**Overall Recommendation:** 4
**Confidence:** 4

**Summary:**

This paper focuses on the online differential privacy query-answering system and proposes the LAPRAS framework, which performs offline matrix mechanism precomputation using prediction set and allocates the remaining privacy budget to unpredicted queries through an adaptive online policy.

**Compliance With Llm Reviewing Policy:**

Affirmed.

**Final Justification:**

The authors have made considerable efforts and addressed most of the concerns raised. Therefore, I am willing to adjust my rating to “Weak Accept.”

**Key Questions For Authors:**

1. How is the prediction set P obtained in practice, and how should its quality be evaluated?
2. Can the authors provide a direct experimental comparison between Static Allocation and Smooth Allocation under the same settings?
3. When the prediction set P increases, particularly when P contains a large number of irrelevant queries that never appear in the actual data stream, does this method incur significant storage and preprocessing overhead, even though the overlap with the actual workload may improve?
4. Could the authors provide additional evaluation metrics beyond Median MAE?

**Limitations:**

No. This paper would be more comprehensive if it more clearly discusses its primary limitations, particularly its dependence on a high-quality prediction set P, the assumption of random order, and the potential storage overhead when P is large.

**Strengths And Weaknesses:**

Strength:
1. The methodological framework of this paper is clearly structured, and the overall design logic is well-defined, making it easy to understand.
2. The paper provides a relatively comprehensive theoretical analysis.
3. The experimental results provide a good support for the core findings the paper aims to highlight.

Weakness：
1. The LAPRAS framework proposed in this paper relies on a high-quality prediction set P. Definition 3.8 specifies that prediction set P is derived from the prior logs or learned workload structure. Furthermore, the experimental results in Figures 1 and 2 demonstrate that the framework's performance is highly sensitive to the quality of P. Consequently, the paper does not elaborate on how to obtain a high-quality P in practical scenarios, leaving this critical prerequisite somewhat idealized.
2. The paper introduces both Static allocation and Smooth allocation methods, but the experimental analysis appears slightly unbalanced: Table 2 presents results for static allocation, and Figures 1, 2 show results for smooth allocation. However, this section lacks a direct side-by-side comparison between the two allocation schemes. Supplementing this comparison would enhance clarity.
3. The experimental results in this paper indicate that at high overlap, the proposed LAPRAS framework achieves performance comparable to OfflineMM. However, if the prediction set P is large but contains many queries that ultimately do not appear, could the preprocessing of offline MM be hindered by irrelevant queries? This point has not been specifically addressed.
4. Section 4.1.1 states that the Matrix Mechanism is applied to the prediction set P and the precomputed answers are stored, but it does not specify the storage overhead of the precomputed results. When P is large, it remains unclear whether the approach remains practical.
5. Theorem 4.2 establishes an unbiased estimator under the assumption that queries arrive in random order, while Smooth Allocation further relies on this estimation result for budget allocation. Considering that real query streams often exhibit burstiness, clustering, or temporal dependencies, it is recommended that the authors supplement their analysis with an examination of this method in non-random arrival scenarios.
6. The current experimental section primarily presents experimental results using Median MAE as the evaluation metric. It is recommended that the authors further supplement the evaluation of method performance with additional metrics to make the experimental conclusions more comprehensive.

---

> ### Author Rebuttal · Authors · 2026-03-31
>
> We thank the reviewer for the constructive feedback and provide additional clarification and results below.
>
> ## How is the predicted set $P$ obtained in practice?
>
> Our intent is that $P$ is obtained from prior logs, recurring workload templates, or any learned predictor over future query demand; LAPRAS does not require a specific prediction mechanism. In the current paper, we abstract this into a predicted set so that we can isolate the core question: how much can one gain in online DP query answering when some fraction of future demand is predictable? We agree the current presentation idealizes this component, and we will clarify this limitation and discuss practical ways to construct and evaluate $P$ more explicitly. In our experiments, prediction quality is controlled through overlap, which serves as a direct proxy for how useful $P$ is.
>
> ## Direct comparison between Static and Smooth
>
> We ran a direct comparison under identical settings (same $\epsilon, \delta$, dataset, $P, S$, overlap, and budget split). Smooth matches Static at overlap = 1.0 by construction, and *progressively outperforms Static once overlap is sufficiently high*. For example, with matrix-heavy, $|P|=50$, $S=50$, Smooth improves MAE over Static by 10.0% at overlap 0.5, 28.8% at 0.6, 38.4% at 0.7, 38.2% at 0.8, and 40.2% at 0.9. Under query-heavy, the same pattern holds, with improvements of 6.9%, 9.0%, 20.3%, 4.3%, and 16.5% over the same overlap range. The crossover point is stable: for $|P|=50$, Smooth first beats Static at overlap $\approx 0.5$ across all four budget strategies; for $|P|=100$, the crossover shifts later to 0.6–0.7, reflecting the effect of more false positives in $P$. The pattern is also consistent across RMSE, NMAE, MAPE, SMAPE, and $R^2$.
>
> ## Effect of large $P$, false positives, and overhead
>
> We additionally varied $|P| \in \{50, 100, 200, 500\}$ and $S \in \{50, 100, 500\}$, so $|P|/S$ ranges from 0.5× to 5×. The main empirical finding is that *overlap is the dominant factor*, while $|P|$ is secondary. Increasing overlap from 0% to 50% reduces Smooth MAE from 304.3 to 116.3 for $|P|=50$, and from 325.7 to 107.0 for $|P|=100$. By contrast, increasing $|P|$ at fixed overlap changes utility only modestly, even when many elements of $P$ never appear. At 100% overlap with $|P|=100$, half of $P$ is still unused, yet Smooth MAE is 72.4 versus 74.6 for $|P|=50$. The main cost of larger $P$ is therefore preprocessing, not a sharp utility collapse. Concretely, the storage overhead is simply the size of the predicted set together with its precomputed answers, and runtime is dominated by the offline MM step: for $S=500$, MM runtime grows from 185.8s ($|P|=50$) to 231.7s ($|P|=100$), 356.9s ($|P|=200$), and 874.6s ($|P|=500$), while Smooth allocation itself remains negligible (<0.2s). We will state this limitation more explicitly.
>
> ## Additional metrics beyond median MAE
>
> We agree that scale-free metrics improve interpretability, and we can add them in the revision. We computed NMAE, range-normalized NRMSE, MAPE, and SMAPE, and they all show the same qualitative trend as MAE: overlap matters much more than $|P|$, and utility improves steadily as prediction quality improves. For example, for Smooth in the query-heavy setting, NMAE drops from 1.14 to 0.355 to 0.315 for $|P|=50$ as overlap goes from 0% to 50% to 100%, and from 1.406 to 0.323 to 0.210 for $|P|=100$.
>
> ## Random-order assumption and empirical scope
>
> We also agree that extending the estimator beyond random-order streams is an important direction, and we discuss this as a limitation of the current work. As an initial check, we additionally evaluated a worst-case *bad-first* ordering, where all unpredicted queries arrive before any predicted ones. As expected, this hurts Smooth more than Static, since unpredicted queries are front-loaded before the estimator can benefit from later predicted queries. However, the degradation is *graceful rather than catastrophic*. In the query-heavy setting, the worst Smooth/Static gap is about 1.6×, and both variants still outperform the independent-noise baseline once overlap is moderate. In the matrix-heavy setting, Smooth is competitive with, and sometimes better than, Static at low overlap because the larger MM allocation provides a stable base even under adversarial ordering. At overlap = 1.0, Static and Smooth coincide by construction. More broadly, LAPRAS is a *framework*: in this paper we instantiate the offline component with the matrix mechanism, but it can be replaced by other workload-aware private solvers.

---

> > ### Author Rebuttal · Reviewer_emXy · 2026-04-05
> >
> > The authors have made considerable efforts and addressed most of the concerns raised. Therefore, I am willing to adjust my rating to “Weak Accept.”

---

> > > ### Author Response · Authors · 2026-04-08
> > >
> > > We are grateful to the reviewer for their engagement with us throughout the review process, their positive assessment of our work, their helpful suggestions, and for raising their score to a Weak Accept. We will be sure to include the direct side-by-side comparisons, the additional relative error metrics, the effect of large $P$, false positives, and overhead, and the expanded discussion of the random order assumption and empirical scope in the final version of our paper.

---

### Official Review · Reviewer_ixmx · 2026-03-13

**Soundness:** 2
**Presentation:** 2
**Significance:** 2
**Originality:** 2
**Overall Recommendation:** 4
**Confidence:** 2

**Summary:**

The paper focus on online differentially private (DP) answering of a random order stream of linear queries under a single global budget, assuming access to a predictor that outputs a set P of queries. The proposed mechanism LAPRAS, precomputes a DP release for predicted queries using the matrix mechanism, and answers the remaining “bad” queries online using a residual budget paced by a "Smooth Allocation scheme" that relies on an unbiased estimator for the number of bad queries. Authors provide proofs of unbiasedness/concentration for the estimator and empirical results on two datasets showing strong gains when overlap is high and graceful degradation when overlap is low.

**Compliance With Llm Reviewing Policy:**

Affirmed.

**Final Justification:**

While I retain practical reservations regarding scalability (the Matrix Mechanism's SDP overhead limits domains to 100 bins) and the use of synthetic overlap rather than real-world query logs, the core idea is compelling as a proof-of-concept.

I am raising my score to a 4 (Weak Accept). I hope the final revision includes the comparisons, relative error metrics, and discussion of the scaling bottlenecks.

**Key Questions For Authors:**

- What is the size of the predicted set P relative to S in your experiments? How sensitive are results to false positives in P (predicted queries that never arrive) and to very large |P|, both in utility (over-spending on MM) and runtime (SDP scalability)?

- How robust is smooth allocation to violations of the random order assumption (e.g. clusters of bad queries early or late)? Can you report experiments with adversarial or semi-adversarial arrival patterns?

- Why choose δ = 1e−3, and what are the dataset sizes N and typical count scales for adult and Gowalla histograms? Could you add relative error or normalized error metrics to ease interpretation across settings?

- Could you compare at least in a controlled setting, against a caching based approach (e.g., CacheDP) under a fixed global budget metric to shows the possible trade-offs?

**Limitations:**

I encourage the authors to revise the theory, expand empirical validation, and clarify related-work positioning.

**Strengths And Weaknesses:**

- A learning augmented point of view to bridge offline and online (streaming) DP.
-It introduces a stopping time estimator for the number of unpredicted queries in a random order stream and integrates it into a budget pacing policy, the smooth allocation.
-The trade-off: substantial error reductions when predictions are accurate and competitive performance with a simple online baseline when predictions are poor.
- Includes both static and smooth allocations, showing that smoothing improves robustness to early-stream anomalies.
- It could contribute for an important setting in DP, online answering with recurring workloads and a single global budget.

Some weakness in my opinion:
- Theorem 4.7 utility bound seems unjustified to me, it mixes incompatible sensitivity and composition models. The proof assumes a joint vector release with $L_2$ sensitivity, but the algorithm actually adds independent online gaussian noise with per-query $\epsilon_i$. Under basic composition, this yields an expected squared error of $\Theta(B^3/\epsilon^2)$. The claimed $\mathcal{O}(B^2 \ln(1/\delta)/\epsilon^2)$ bound sounds invalid without advanced accounting (e.g., zCDP/RDP) or a true joint release.

- The privacy proof relies on basic composition and splitting. However, the utility analysis implicitly assumes advantages that typically require advanced composition or concentrated DP accounting, which is not reflected in the algorithm or the proof.

From an empirical prospective:
- Evaluation uses small stream sizes and only two datasets.  The results lack statistical depth for a systems-style claiming.
- Baselines are limited to independent per query noise and an offline matrix mechanism. While the paper argues CacheDP is orthogonal, it is still a practical online DP system; a qualitative or quantitative comparison (e.g., measuring end utility under a shared global budget rather than accuracy constraints) would be great.
- The overlap model is synthetically controlled. There is no study of miscalibrated predictions beyond overlap rate ( P size, false positives that never arrive, structure mismatch), or robustness to violations of the random-order assumption.
- The choice δ = 1e−3 and the lack of dataset size N/reporting of sensitivity magnitudes make absolute AMAE hard to interpret across scenarios.

Honestly, my review confidence is gonna be low and I can assume that I didn't get the paper  fully. But in my opinion, as written, the utility bound and the constant factor comparison to the baseline seems not justified and make the  main contribution (theory)  not strong. The experiments, remain limited (small S, few datasets, few runs, synthetic overlap), and no realistic workloads, robustness to violations, and broader SoTA baselines.

---

> ### Author Rebuttal · Authors · 2026-03-31
>
> We thank the reviewer for the detailed questions.
>
> ## Utility Proof Advanced Composition
>
> Under the basic composition, splitting the budget across $B$ queries yields an expected squared error that scales with $O(B^3/\varepsilon^2)$, not $O(B^2/\varepsilon^2)$. Adaptive advanced composition can be used to show the improved result given in Theorem 4.7 since we use the Gaussian mechanism with $(\varepsilon, \delta)$-DP. However, for simplicity, we will correct Theorem 4.7 to explicitly state the $O(B^3/\varepsilon^2)$ bound using online, basic-composition.
>
> **Theoretical contribution:** Our primary theoretical contribution is the non-trivial unbiased stopping-time estimator for bad queries (Theorem 4.2) and its variance bound (Corollary 4.4), which use a novel Negative Hypergeometric analysis and are crucial components of our algorithm.
>
> ## Sensitivity to $|P|$, false positives, and runtime
>
> We explicitly define the *false-positive rate* of the predicted set $P$ as the fraction of queries in $P$ that never appear in the realized stream. If the stream contains $S$ queries and the overlap fraction is $\alpha$, then the number of true positives is $\alpha S$, the number of false positives is $\max(0, |P|-\alpha S)$, and the false-positive rate is $\max(0, |P|-\alpha S)/|P|$. We varied $|P| \in \{50, 100, 200, 500\}$ and $S \in \{50, 100, 500\}$, so $|P|/S$ ranges from 0.5× to 5×. This lets us directly study both larger predicted sets and higher false-positive rates.
>
> The main finding is that *overlap is the dominant factor*, while $|P|$ and the induced false-positive rate are secondary. Under query-heavy settings, increasing overlap from 0% to 50% reduces Smooth MAE from 304.3 to 116.3 for $|P|=50$, and from 325.7 to 107.0 for $|P|=100$. At fixed overlap, increasing $|P|$ changes utility only modestly, even when many elements of $P$ are false positives. For example, at overlap = 1.0 with $|P|=100$ and $Q=50$, only 50 of the 100 predicted queries appear, so the false-positive rate is 50%; yet Smooth MAE is 72.4, very close to 74.6 for $|P|=50$, where the false-positive rate is 0%. This suggests that false positives do not materially hurt utility in our setting, although very large $P$ increases preprocessing cost. Runtime is dominated by the offline MM step: for $Q=500$, MM runtime grows from 185.8s ($|P|=50$) to 231.7s ($|P|=100$), 356.9s ($|P|=200$), and 874.6s ($|P|=500$), while Smooth allocation itself remains negligible (<0.2s).
>
> ## Robustness to adversarial arrival order
>
> We additionally evaluated a worst-case *bad-first* ordering, where all unpredicted queries arrive before any predicted ones. As expected, this hurts Smooth more than Static, since unpredicted queries are front-loaded before the estimator can benefit from later predicted queries. However, the degradation is *graceful rather than catastrophic*. In query-heavy settings, the worst Smooth/Static gap is about 1.6×, and both variants still substantially outperform the independent-noise baseline once overlap is moderate.
>
> ## Relative / normalized error metrics
>
> We report these additional metrics with $\delta=10^{-10}$, and the qualitative trends are consistent with those in the paper. We computed NMAE, range-normalized NRMSE, MAPE, and SMAPE, and they all show the same pattern as MAE: *overlap matters much more than $|P|$*, and utility improves steadily as prediction quality improves. For Smooth in the query-heavy setting, NMAE drops from 1.14 to 0.355 to 0.315 for $|P|=50$ as overlap goes from 0% to 50% to 100%, and from 1.406 to 0.323 to 0.210 for $|P|=100$.
>
> Regarding dataset size, our current experiments use histograms with 100 bins because the Matrix Mechanism requires solving an SDP, and this becomes the computational bottleneck as the domain size grows. Since LAPRAS is a framework, our goal is to validate the learning-augmented effect in a controlled setting.
>
> ## Controlled comparison with CacheDP
>
> We also ran a controlled comparison against CacheDP on Adult using the *same total privacy budget*. To make the comparison favorable to CacheDP, we first issue the predicted-set queries to warm the cache, and then answer the stream. Even in this setting, LAPRAS is substantially more accurate under normalized metrics. For overlap 0 / 0.5 / 1.0, Static NMAE is 0.0019 / 0.0007 / 0.0014, whereas CacheDP gives 0.2076 / 0.1242 / 0.0588; similar gaps hold for NRMSE. LAPRAS uses an optimized offline mechanism on the prediction, whereas CacheDP spreads the same global budget across many online answers.
>
> ## Empirical evaluation
>
> Our current experiments are proofs of concept *focused on accuracy* under controlled overlap and ordering conditions, rather than as a full systems-scale study. More broadly, *LAPRAS is a framework*: in this paper we instantiate the offline component with the matrix mechanism, but that component can be replaced by other workload-aware private solvers.

---

> > ### Author Rebuttal · Reviewer_ixmx · 2026-04-03
> >
> > Thank you for the rebuttal. While I retain practical reservations regarding scalability (the Matrix Mechanism's SDP overhead limits domains to 100 bins) and the use of synthetic overlap rather than real-world query logs, the core idea is compelling as a proof-of-concept.
> >
> > I am raising my score to a 4 (Weak Accept). Please ensure the final revision includes the comparisons, relative error metrics, and discussion of the scaling bottlenecks.

---

> > > ### Author Response · Authors · 2026-04-08
> > >
> > > We are grateful to the reviewer for their time in engaging with us throughout the review process, their positive assessment of our work, their helpful suggestions, and for raising their score to a 4. We will be sure to include the comparisons, relative error metrics, and discussion of the scaling bottlenecks in the final version of our paper.

---

### Decision · Program_Chairs · 2026-04-30

**Decision:**

Accept (regular)

**Comment:**

This paper studies online differentially private answering of streaming linear queries and proposes LAPRAS, a learning-augmented framework that uses a predicted query set to combine an offline matrix mechanism with adaptive online budget allocation. Reviewers agreed that the problem is important and that the learning-augmented perspective is interesting, with particular strength in the stopping-time estimator and the overall consistency-versus-robustness framing.

After considering the reviews, rebuttal, and discussion, I find the paper strong enough for acceptance. The rebuttal substantially strengthened the submission by addressing the main empirical concerns with additional results on direct Static versus Smooth comparisons, false positives and large predicted sets, adversarial arrival order, controlled comparison against caching-based alternatives, additional normalized metrics, and a more realistic workload derived from IDEBench. These additions helped clarify both when the method provides meaningful gains and how it degrades when predictions are imperfect.

The paper is not without limitations. Reviewers noted the dependence on a useful prediction set, the random-order assumption, and the scalability bottleneck introduced by the matrix mechanism instantiation. One reviewer also remained less convinced about overall significance. However, the core idea is technically sound, the rebuttal resolved most major concerns, and the resulting contribution is strong enough to merit acceptance as a solid contribution to private online query answering.